# Fingerprints of brain disease: connectome identifiability in Alzheimer's disease
Sara Stampacchia [1] ✉, Saina Asadi[2], Szymon Tomczyk[3], Federica Ribaldi[3,4], Max Scheffler[5], Karl-Olof Lövblad[2,6], Michela Pievani[7], Aïda B. Fall[8,9,10], Maria Giulia Preti [1,2,9], Paul G. Unschuld [11,12], Dimitri Van De Ville [1,2], Olaf Blanke [1], Giovanni B. Frisoni[3,4], Valentina Garibotto[2,9,13,16] & Enrico Amico [1,14,15,16] ✉

Functional connectivity patterns in the human brain, like the friction ridges of a fingerprint, can uniquely identify individuals. Does this "brain fingerprint" remain distinct even during Alzheimer's disease (AD)? Using fMRI data from healthy and pathological ageing subjects, we find that individual functional connectivity profiles remain unique and highly heterogeneous during mild cognitive impairment and AD. However, the patterns that make individuals identifiable change with disease progression, revealing a reconfiguration of the brain fingerprint. Notably, connectivity shifts towards functional system connections in AD and lower-order cognitive functions in early disease stages. These findings emphasize the importance of focusing on individual variability rather than group differences in AD studies. Individual functional connectomes could be instrumental in creating personalized models of AD progression, predicting disease course, and optimizing treatments, paving the way for personalized medicine in AD management.

The sharp increase of publicly available neuroimaging datasets[1–3] in the last few years has provided an ideal benchmark for mapping functional and structural connections in the human brain. At the same time, quantitative analysis of connectivity patterns based on network science has become more commonly used to study the brain as a network[4,5], giving rise to the scientific field of brain connectomics[6]. Seminal work in this research area[7–9] has paved the way towards the new promising avenue of detecting individual differences through brain connectivity features. These studies showed that an individual's functional connectivity patterns estimated from functional magnetic resonance imaging (fMRI) data, also known as functional connectomes (FCs), can constitute a marker of human uniqueness, as they can be used to identify a given individual in a set of functional connectivity profiles from a population[9]. Given the analogy to well-known properties of the papillary ridges of the human finger, the field has taken the name of 'brain fingerprinting' and, since then, the extraction of "fingerprints" from

human brain connectivity data has become a new frontier in neuroscience, well beyond fMRI data. In fact, studies have investigated brain fingerprints in electroencephalogram (EEG[10–12]) or functional near-infrared spectroscopy (fNIRS[13]), and very recently from magnetoencephalography (MEG) in order to investigate neural features of individual differentiation[14–18]. The excitement produced by the discovery that brain fingerprints can be extracted from matrices summarising human brain activity, either during rest or when performing a task, is unsurprising, for several reasons. Firstly, it confirms that studying the brain as a network can provide useful tools to get insights into the individual features that distinguish our brains one from the other; and second, it has been shown that brain fingerprints relate to behavioural and demographic scores[19,20]. Accordingly, efforts have been made to implement ways of maximising and denoising fingerprints from brain data[21–23]. These findings incentivized human neuroimaging studies to advance from inferences at the population level to the single-subject level

[1]Neuro-X Institute and Brain Mind Institute (BMI), École Polytechnique Fédérale de Lausanne (EPFL), Geneva, Switzerland. [2]Department of Radiology and Medical Informatics, Geneva University Neurocenter, University of Geneva, Geneva, Switzerland. [3]Laboratory of Neuroimaging of Aging (LANVIE), University of Geneva, Geneva, Switzerland. [4]Geneva Memory Center, Department of Rehabilitation and Geriatrics, Geneva University Hospitals, Geneva, Switzerland. [5]Division of Radiology, Geneva University Hospitals, Geneva, Switzerland. [6]Neurodiagnostic and Neurointerventional Division, Geneva University Hospitals, Geneva, Switzerland. [7]Lab of Alzheimer's Neuroimaging and Epidemiology, IRCCS Istituto Centro San Giovanni di Dio Fatebenefratelli, Brescia, Italy. [8]Faculty of Medicine, University of Geneva, Geneva, Switzerland. [9]CIBM Center for Biomedical Imaging, Lausanne, Switzerland. [10]Department of Psychiatry, Geneva University Hospitals, Geneva, Switzerland. [11]Division of Geriatric Psychiatry, University Hospitals of Geneva (HUG), 1226 Thônex, Switzerland. [12]Department of Psychiatry, University of Geneva (UniGE), 1205 Geneva, Switzerland. [13]Division of Nuclear Medicine and Molecular Imaging, Geneva University Hospitals, Geneva, Switzerland. [14]School of Mathematics, University of Birmingham, Birmingham, UK. [15]Centre for Human Brain Health, University of Birmingham, Birmingham, UK. [16]These authors contributed equally: Valentina Garibotto, Enrico Amico. ✉e-mail: sara.stampacchia@epfl.ch; e.amico@bham.ac.uk

and allowed the field to move towards an individualised prediction of cognition and behaviour from brain connectomes[24–27]. The next natural step is to explore whether this property of the human brain is maintained during disease. Despite promising preliminary findings towards this direction[16,28], it is to date unclear to what extent FC fingerprints could be used for mapping disease from human brain data. In fact, the vast majority of studies on brain fingerprinting have focused on healthy individuals, leaving the field of brain fingerprinting within the context of brain diseases largely, if not completely, unexplored.

Cognitive decline and dementia are the final consequences of a series of pathological brain events. In the case of Alzheimer's disease (AD), which is the most common cause of dementia, these events involve the accumulation of toxic proteins such as β-amyloid and hyperphosphorylated tau between and within neurons, leading to neuronal death and ultimately causing damage to the wider structural and functional architecture[29]. In line with this, AD is often referred to as a 'disconnection syndrome'[30] and, numerous studies have focussed on connectivity alterations in AD[31–33]. This extensive body of literature has demonstrated that AD is associated with loss of functional connectivity between brain regions and disruption of network organisation. Overall, the literature suggests that during AD, the brain undergoes both a loss and a reorganization of functional connectivity,[31–44] and that these changes are closely tied to the underlying β-amyloid[45,46] and tau pathophysiology[47–50].

However, to date, there is a lack of a functional connectivity alteration biomarker with an adequate level of specificity and sensitivity concerning the various stages of AD[32], and this has hindered the routine use of fMRI in clinical practice[39,51]. This is partly due to the intrinsic properties of connectivity from resting-state fMRI, a technique that is greatly influenced by factors affecting the obtained signal (heterogeneity in acquisition parameters, scanners characteristics, and motion[52]), by differences in the chosen signal-processing approaches[39], but also by the fact that existing studies focused on group averages, overlooking heterogeneity among individuals. Addressing the open research question of fingerprinting during brain disease could therefore open the door to individual characterization of AD from functional connectivity data and pave the way to a more widespread implementation in clinical settings. Investigating fingerprints of stages of AD is also tightly connected to the concept of "precision medicine"[53], since it might not only provide insights on the individual trajectories of pathological brain ageing, but also allow for the surveillance of adapted personalised treatments during AD, advancing medicine in its quest for individualised biomarkers of neurodegeneration[54].

In this work, we investigated within-scan brain connectivity fingerprints using fMRI data collected from two independent datasets from healthy ageing individuals and patients at different stages of cognitive decline due to AD. This study builds upon prior research on brain fingerprinting in AD[16] by pioneering four significant advancements: (a) employing fMRI, replacing the previously utilized MEG technology; (b) broadening the scope of the investigation to encompass the advanced stages of the disease, including dementia in addition to mild cognitive impairment (MCI); (c) exclusively enroling amyloid-positive patients, thereby ensuring a focus on cognitive decline specifically attributable to AD; and (d) integrating data from two distinct cohorts, enhancing the robustness and generalizability of the findings. We started by estimating functional connectome fingerprints of a clinically homogeneous and deeply characterised cohort of cognitively unimpaired amyloid β-negative individuals (CU Aβ−), amyloid β-positive patients with mild cognitive impairment (MCI Aβ+) and amyloid β-positive patients with dementia due to AD, during the first and second half of the fMRI session. We found that whole-brain functional connectivity patterns remained reliable across healthy and pathological brain ageing; in other words, it was possible to correctly identify a patient solely based on its functional connectome. Yet, significant differences in the spatial organisation of the brain fingerprint could be observed during the different stages of AD. Notably, the functional connections that were the most reliable in CU Aβ− disappeared in MCI Aβ+ and AD dementia, leaving room for other stable connections, adjusted

to the process of neurodegeneration. Furthermore, we looked at the distribution of connections with the highest fingerprint (i.e. temporally stable across time and allowing subject identification), and we found a significant transition towards between-functional system connections in key functional-networks, when going from healthy to AD. Finally, this topological reconfiguration appeared to be associated with mostly high-order cognitive processes in CU Aβ−, and shifted to other functions during the different stages of AD. These findings emphasize the significance of harnessing brain uniqueness by shifting the research focus from group differences to individual variability when investigating functional connectivity alterations in AD.

## Results

We investigated within-session brain fingerprints of neurodegeneration in two independent cohorts for a total of $N = 126$ subjects (i.e. Geneva cohort: $N = 54$ and ADNI: $N = 72$) during healthy ageing and different stages of AD: CU Aβ−, MCI Aβ+ and AD dementia. Our approach can be summarised in three steps: (1) we first estimated the FCs of each subject during the first and second halves volumes of fMRI acquisitions separately, (cf. Fig. 1A and see "Methods" for details). (2) We then estimated the degree of within-session brain identification or "brain fingerprint" at the whole-brain level for each group separately, through a mathematical object called identifiability matrix[22] (cf. Fig. 1B). The identifiability matrix provides two useful metrics for brain fingerprinting: the degree of similarity of each subject with itself (ISelf, diagonal elements, Fig. 1B) as opposed to others (IOthers, off-diagonal elements Fig. 1B), and the degree of brain discriminability, conceptualised as the extent to which subjects were more similar to themselves than others (IDiff, see "Methods"). We also estimated the success rate of the identification procedure, as originally proposed by ref. 9. (3) We further explored the spatial specificity of brain fingerprints by estimating the degree of distinctiveness of each FC-edge at the individual level, using intraclass correlation (ICC, see "Methods" and Fig. 1C). Edge-wise fingerprint was then computed at the networks (Fig. 1C) and hubs level (Fig. 1D).

### Whole-brain within-sessions brain fingerprint during AD

In these first analyses, we investigated the degree of identification at the whole-brain level. We found that the success rate of the identification procedure was 100% in both the Geneva and ADNI cohorts. In other words, in each group, each individual showed significantly higher similarity with themselves (ISelf), as opposed to others (IOthers) (Fig. 2B, D, Geneva and ADNI $p < 0.0001$), and IDiff was high in the three groups. We also found that test–retest reliability (ISelf) was high in the three groups and in both cohorts, with no significant differences across groups after controlling for nuisance variables, i.e. age, sex, years of education (YoE) and absolute difference between motion (FD) at test vs retest [ANOVA with 5000 permutations to control for sample size differences; Geneva: F(2) = 0.08, $p = 0.918$; CU Aβ−: $N = 16$; M(SD) = 0.60(0.07); MCI Aβ+: $N = 32$; M(SD) = 0.60(0.10); AD dementia: $N = 6$; M(SD) = 0.60(0.07); cf. Fig. 2A; ADNI: F(2) = 0.95, $p = 0.389$; CU Aβ−: $N = 40$; M(SD) = 0.73(0.07); MCI Aβ+: $N = 21$; M(SD) = 0.70(0.08); AD dementia: $N = 11$; M(SD) = 0.71(0.08); cf. Fig. 2C]. See Supplementary Fig. 1A for boxplots of ISelf across groups and Supplementary Table 1A for full statistics for one-way ANOVAs. We tested differences in between-subjects similarity (IOthers) after controlling for nuisance variables (i.e. age, sex, YoE, average motion (FD) and scanner type, the latter for the ADNI dataset only). We found a main effect of the group [ANOVA with 5000 permutations to control for sample size differences; Geneva: $N = 54$; F(2) = 15.3, $p < 0.001$; ADNI: $N = 72$; F(2) = 4.2, $p = 0.021$], revealing reduced between subjects similarity in MCI Aβ+ patients relative to CU Aβ− in the Geneva cohort [Bonferroni adjusted; MCI Aβ+ vs CU Aβ−: $p < 0.0001$] and in AD dementia patients relative to CU Aβ− in ADNI [Bonferroni adjusted; ADNI: AD dementia vs CU Aβ−: $p = 0.009$]. See Supplementary Fig. 1B for boxplots of IOthers across groups and Supplementary Table 1B, C for full statistics for one-way ANOVAs and post-hoc pairwise comparisons. Finally, permutation testing showed that IDiff and Success rates were different from null distributions at $p < 0.01$ in all groups

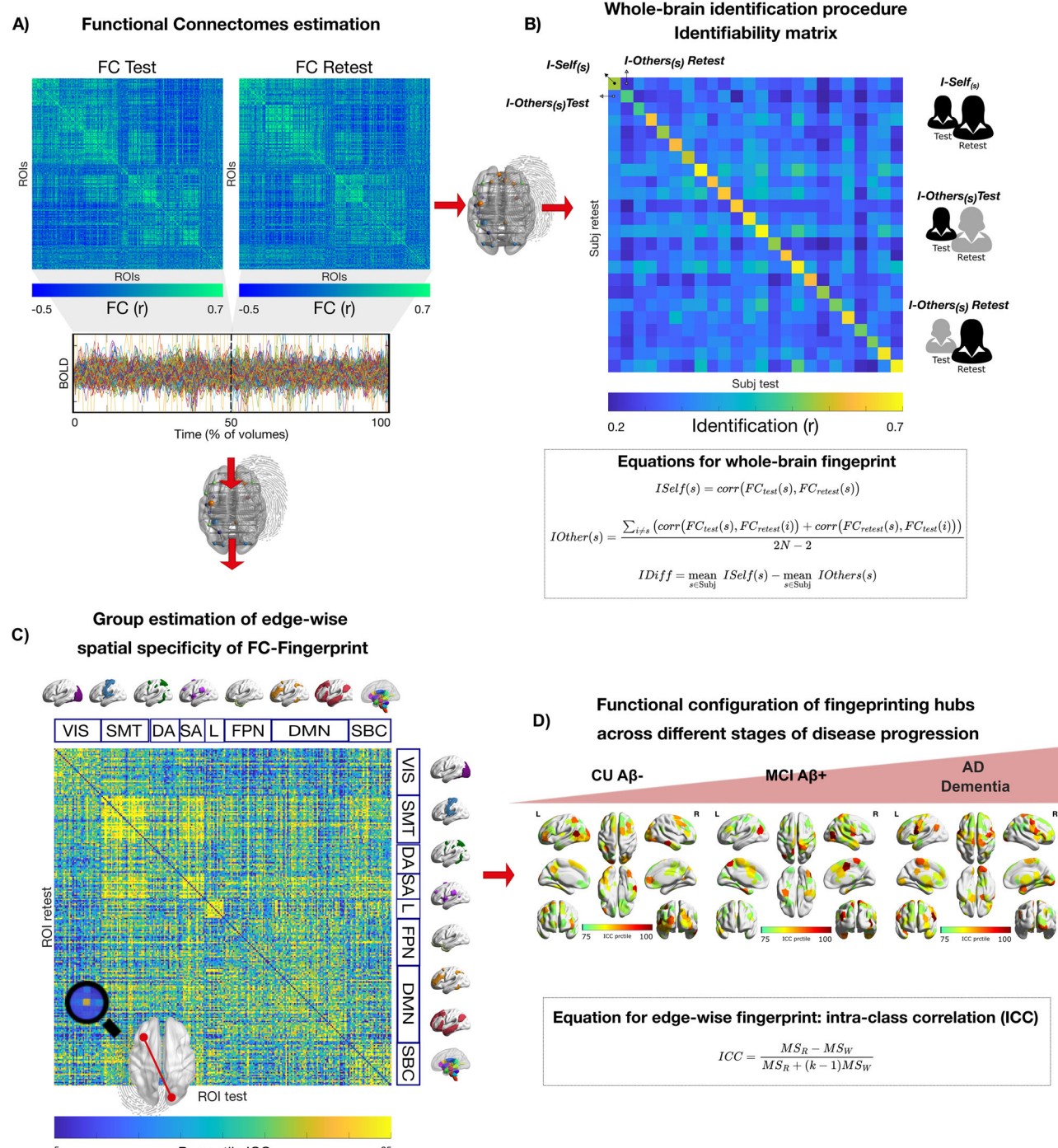

**Fig. 1 | Exploring FC fingerprints of neurodegeneration, schematics of the approach.** We estimated the FC fingerprint in cognitively unimpaired Aβ-negative (CU Aβ−), MCI Aβ+ and dementia Aβ+ (AD dementia) patients across two independent cohorts (Geneva, ADNI). The within-session fingerprint was estimated at the whole brain level as the degree of similarity between functional connectivity at test (FC Test, first 50% volumes) vs retest (FC Retest, second 50% volumes, see "Methods" for details) (**A**) and summarised in a mathematical object called identifiability matrix (**B**)[22]. The identifiability matrix shows within-subjects similarity (*ISelf*, elements in diagonal) and between-subjects similarity (*IOthers*, off-diagonal elements) across each group/cohort. Where *ISelf* > *IOthers* the identification procedure is successful. We also estimated *IDiff*, an estimation of the group-level whole-brain fingerprint as the distance between *ISelf* and *IOthers* (see "Methods"). **C** Spatial specificity of FC fingerprint was estimated for each group/cohort using ICC, quantifying the fingerprint for each brain edge (connection). ICC matrix is ordered according to the seven cortical resting state networks (RSNs) as proposed by ref. 91. **D** Fingerprinting hubs were computed as nodal strength of the ICC matrix for each group. VIS visual network, SMT somatomotor network, DA dorsal-attention network, SA salience network, L limbic network, FPN fronto-parietal network, DMN default-mode network, SBC subcortical regions. Icons were downloaded from https://thenounproject.com/ under a Creative Commons Attribution License (CC BY 3.0).

**Fig. 2 | Functional Connectivity fingerprints during AD. A–C** Identifiability matrices show within- (*ISelf*) and between-subjects (*IOthers*) test–retest reliability as Pearson correlation coefficient in CU Aβ− (Geneva *N* = 16; ADNI *N* = 40), MCI Aβ+ (Geneva *N* = 32; ADNI *N* = 21) and AD dementia (Geneva *N* = 6; ADNI *N* = 11), for the two independent cohorts investigated (Geneva *N* = 54 and ADNI *N* = 72, see "Methods" for details). Individuals' *ISelf* and *IOthers* are displayed, respectively, in the diagonal and off-diagonal elements of the matrix. The average *ISelf*, *IDiff* and *Success-rate* were similar in the three groups and *IDiff* and *Success-rate* significantly differed from random distributions. **B–D** Boxplots shows that *ISelf* was significantly higher (paired-sample *t*-test) than *IOthers* in all individual cases and in all groups, for both the Geneva and the ADNI datasets. \*\**p* ≤ 0.01. Error bars indicate the 25th and 75th percentiles.

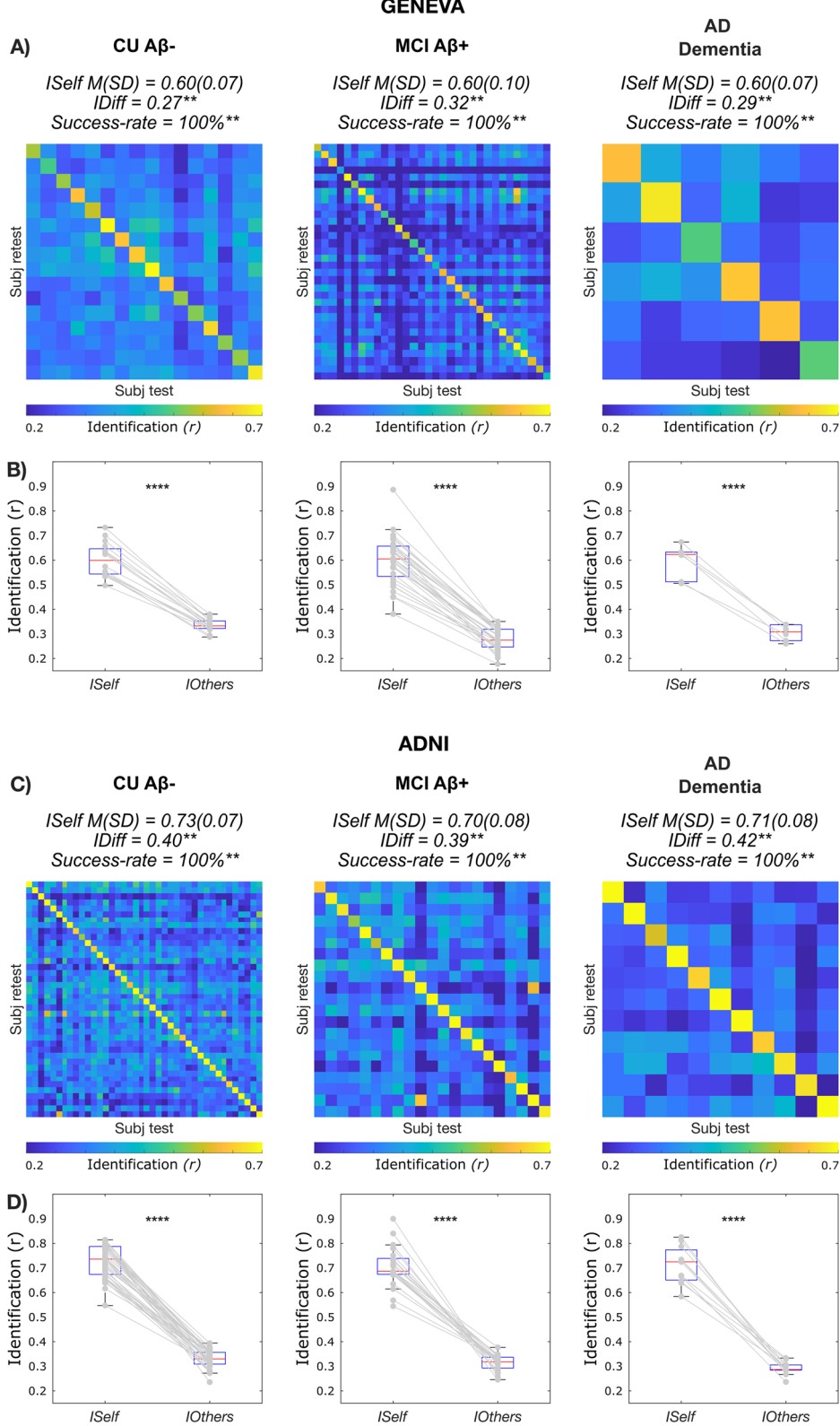

and in both cohorts (Fig. 2A, C). In sum, these data show that it is possible to correctly identify an individual with significantly greater accuracy relative to surrogate null models[15] (see "Methods" for details), independently from the clinical status, and solely based on the patterns of brain activity within the scan. We note that the identifiability results were replicated in two independent cohorts, using two different preprocessing pipelines.

## Spatial specificity of brain fingerprint during AD

We next assessed the spatial specificity of brain fingerprints using edgewise intra-class correlation[22] (ICC). In both cohorts, we observed a spatial reconfiguration of the most identifiable edges as AD progressed (cf. Fig. 3A, B). In other words, connections with the highest ICC had a different topological distribution in the three groups. Although there were differences between the two cohorts, we also identified common features that were

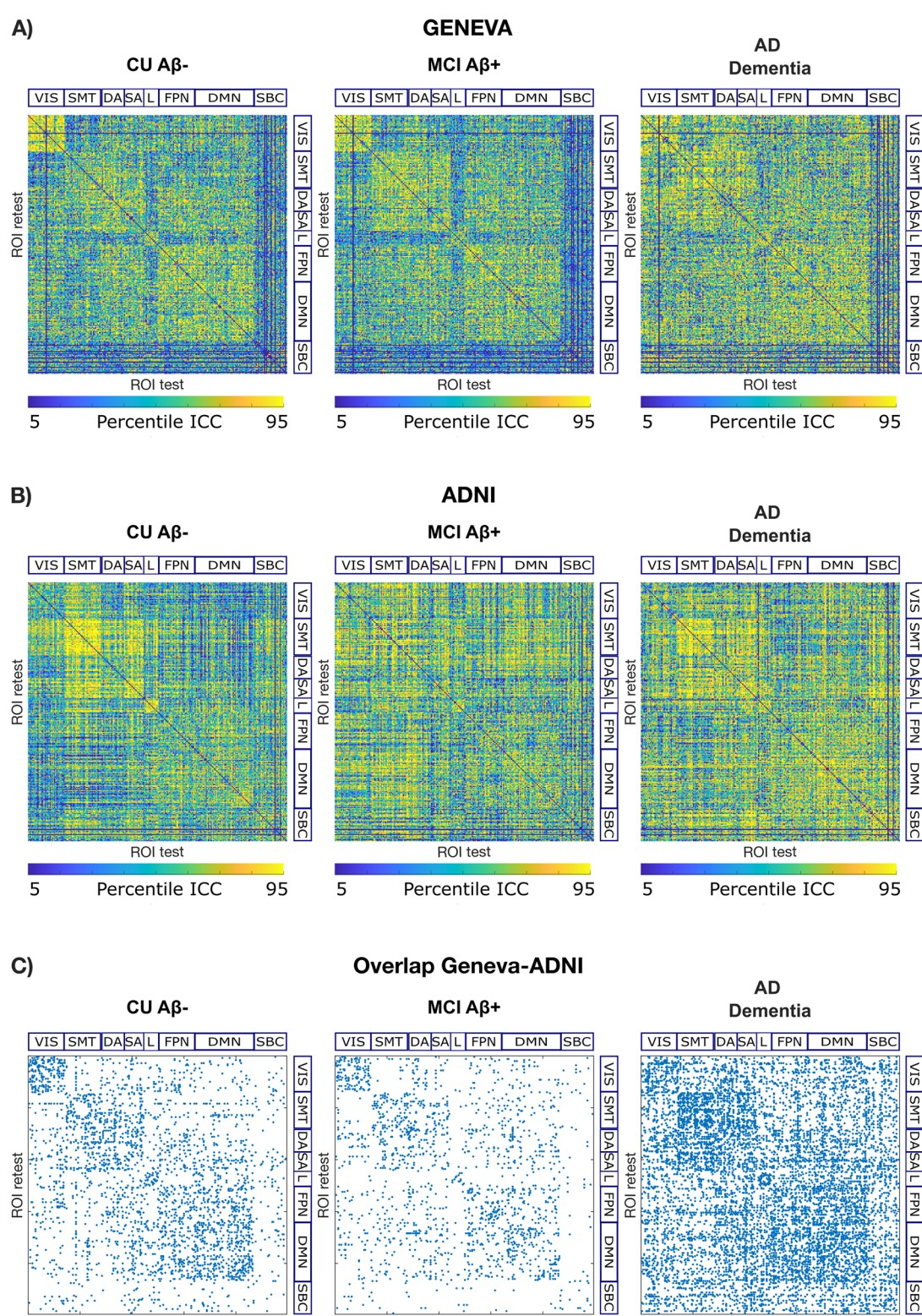

**Fig. 3 | The connections with the highest fingerprint undergo spatial reconfiguration during AD. A** Geneva cohort; **B** ADNI cohort; spatial specificity matrices of FC-fingerprints for each group as measured using edge-wise intra-class correlations (ICC). Here we display ICC values, computed using bootstrapping for a subset of randomly chosen subjects, across 1000 bootstrap runs, and then averaged within each group. We show edges with ICC between the 5th and 95th percentile. **C** Overlap Geneva-ADNI; overlap across the two cohorts of spatial specificity matrices of FC-fingerprints. Matrices in (**A**, **B**) were binarized for ICC > 0.6 which is considered a 'good' ICC score[55], and only overlapping edges are displayed. $n$ edges = number of overlapping edges. Matrices in (**A**–**C**) are ordered according to seven cortical RSNs as proposed by ref. [91]. VIS visual network, SMT somatomotor network, DA dorsal-attention network, SA salience network, L limbic network, FPN fronto-parietal network, DMN default-mode network, SBC subcortical regions.

**Article**

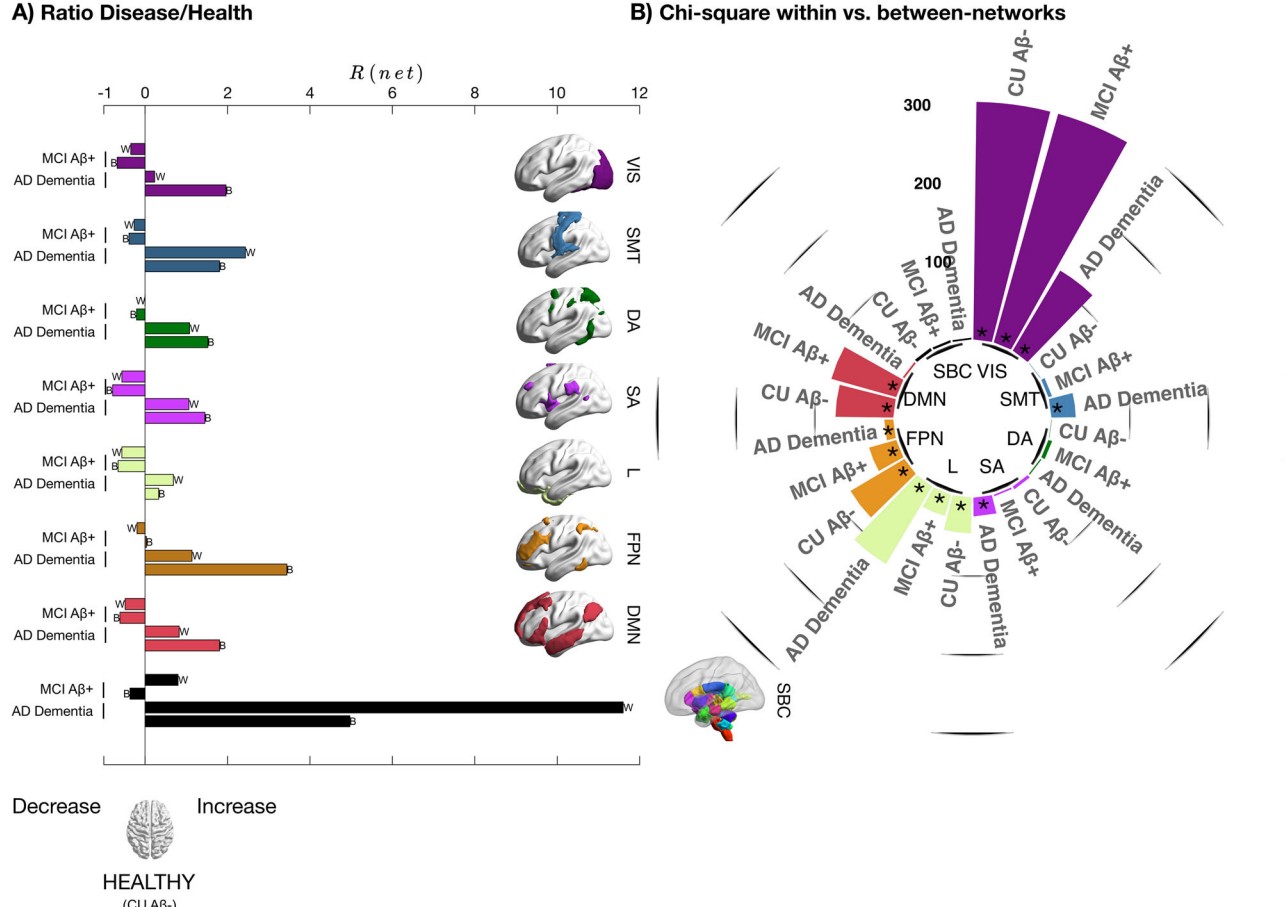

**Fig. 4 | Distribution of brain fingerprint across resting-state functional networks.** Distribution of the edges with the highest ICC common to both cohorts (from ICC overlap binary matrix, cf. Fig. 3C) in within-networks and between-networks. **A** Distance from healthy reference computed as ratio disease/health, R(net), cf. "Methods"). Positive/negative values denote increase/decrease in the percentages of edges, respectively. Note, across all networks, the overall increase in percentages of edges in AD dementia patients and a slight decrease in MCI Aβ+. **B** Comparison of edges within- vs between-networks expressed as Chi-Square statistics. A high value with * denotes a significant (Bonferroni corrected) difference in the number of edges shared. To investigate these common features, we examined the overlap in within-networks vs between-networks. With some exceptions (see main text and Supplementary Fig. 2), this reflects a higher percentage of edges within relative to between-networks. The chi-square for VIS in CU Aβ− and MCI Aβ+ was >800, but here we display a chi-square ≤ 300 for visualisation purposes. W within-networks, B between-networks, ICC intra-class correlation, CU Aβ− cognitively unimpaired Aβ-negative, VIS visual network, SMT somatomotor network, DA dorsal-attention network, SA salience network, L limbic network, FPN fronto-parietal network, DMN default-mode network, SBC subcortical regions.

shared. To investigate these common features, we examined the overlap between the edges with 'good' levels of ICC (ICC > 0.6)[55] across the two cohorts (cf. overlap ICC matrix in Fig. 3C). We observed a slight decrease in the number of edges with 'good' ICC in MCI Aβ+ relative to CU Aβ−, and a large increase in AD dementia relative to both CU Aβ− and MCI Aβ+ (cf. '*n* edges' in Fig. 3C). The lower count of edges with high ICC in the MCI Aβ+ group may reflect the incipient AD pathology causing reorganization, and the fact that MCI Aβ+, though biologically homogeneous, presents with distinct clinical subtypes (e.g. amnestic vs non-amnestic)[56] and uncertain disease trajectories (i.e. not all MCI Aβ+ convert to AD dementia[57,58]). Given these observations, we delved deeper into the spatial distribution of the edges with good ICC within each resting-state network.

**Brain fingerprint in resting-state functional networks during AD**
We looked at the number of edges in the binary overlap ICC matrix for each group (CU Aβ−, MCI Aβ+ and AD dementia) in each network (within and between), relative to the total number of edges. Our results showed that MCI Aβ+ individuals had a slightly reduced fingerprint relative to CU Aβ− (i.e. slightly negative ratio disease/health, R(net)), with an average of −0.2 and −0.5 folds relative to CU Aβ− in within- and between-networks, respectively. In contrast, patients with AD dementia showed a considerable increase in fingerprint (i.e. positive R(net)) across all functional networks

(see Fig. 4A). The greatest increase was in the SBC network, including hippocampal and medial temporal regions, which are the earliest regions displaying atrophy and tau pathology in AD[59,60]. In addition, in between-networks, there was an average increase of 2.4 folds, in particular for DMN, FPN and VIS. There was also a higher fingerprint in within-networks, with an average increase of 2.2 folds. This shows that the increase in a number of edges with the highest ICC in AD dementia was widely distributed across the cortex especially driven by the regions with the most long-lasting neuropathology, and by between-network connections in key resting-state functional networks.

Next, we analysed the differences in within-network and between-networks fingerprints. We conducted a Chi-Square test to compare the number of edges within each network vs the number of edges of that network with the others (i.e. between-network). The results showed that in CU Aβ−, in VIS, Limbic, FPN and DMN the proportion of edges with high fingerprint was significantly higher (Bonferroni corrected) in within-networks relative to between-networks (see Fig. 4B, see also Supplementary Fig. 2). Conversely, in both MCI Aβ+ and AD dementia a notable reduction in Chi-Square statistics was observed in the FPN network. This indicates that, relative to CU Aβ−, patients with AD had higher fingerprints in the connections of the FPN with the rest of the brain. In Visual and DMN networks, a similar reduction in Chi-Square was observed, but only for AD

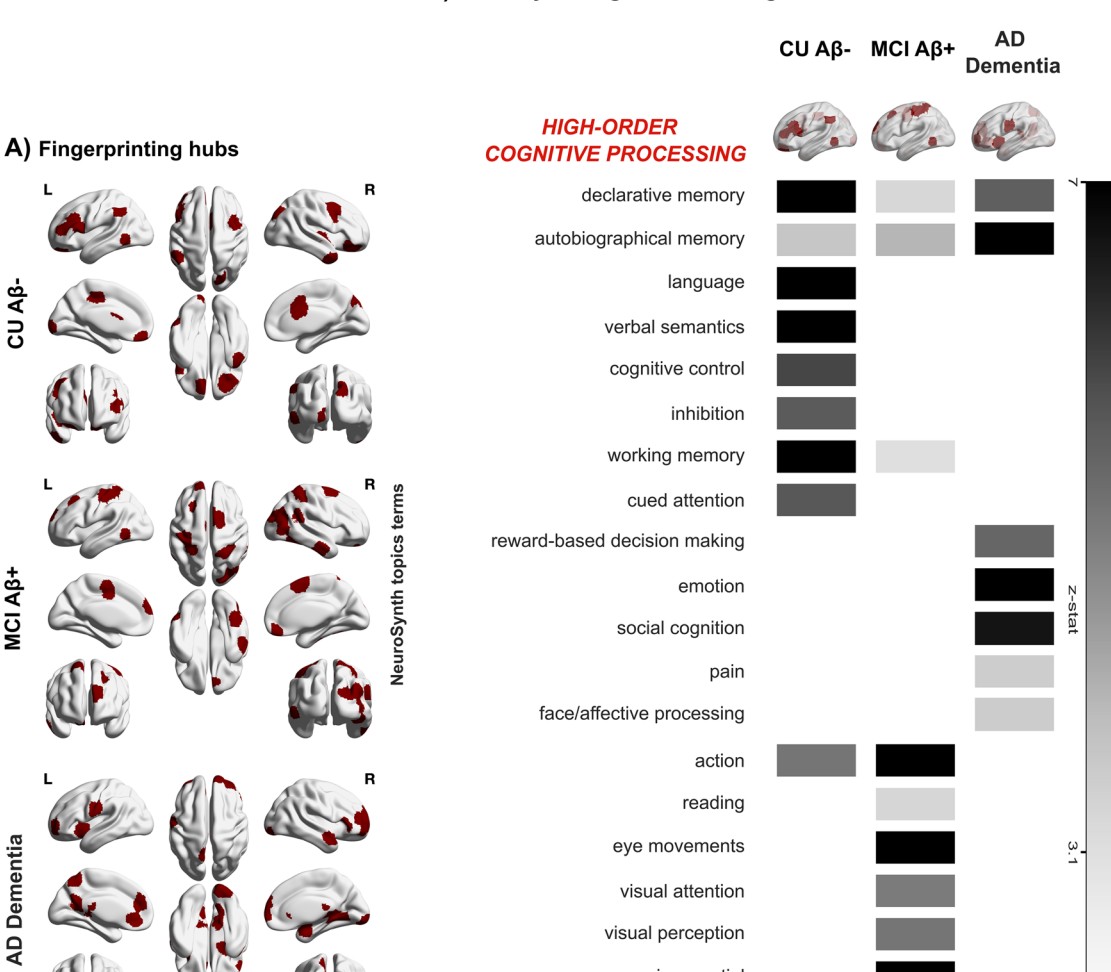

**B) NeuroSynth cognitive decoding**

**A) Fingerprinting hubs**

**Fig. 5 | Fingerprinting *hubs* of AD and its association with behaviour. A** Brain fingerprint map showing the top 25% of brain nodes overlapping across the two cohorts. **B** The Neurosynth meta-analysis of the brain fingerprints maps across healthy ageing and AD show a spectrum of association with higher order processes in CU Aβ−, towards lower-order sensory-motor processing during MCI Aβ+ and back to a higher order, as well as affective, pain and social processing, during AD dementia. Brain fingerprints were linked with memory processes both in healthy ageing and during AD dementia, with different regions driving this association.

dementia, indicating a higher number of edges in between networks only in the advanced stages of the disease. For the Limbic network, we observed a reduction in Chi-Square for MCI Aβ+ and an increase during AD dementia. This reveals a diminished discrepancy in edges with high fingerprint for between vs within-network connections during the intermediate phases of the disease (MCI Aβ+), while during AD dementia the Limbic network had an even higher proportion of edges with high fingerprint in within-networks, relative to between-network connections. Additionally, patients also showed increased within/between-networks proportion difference in other networks, where CU Aβ− did not show any notable difference. For instance, AD dementia had increased fingerprints in within-networks connections in SMT and in between-networks connections in SA networks. See also Supplementary Fig. 2, for the percentages of edges with high fingerprint in each network across groups.

In summary, these findings suggest that the distribution of edges with the highest fingerprint across resting-state functional networks undergoes changes during AD. Specifically, the distribution shifts towards more between-network connections in some key networks such as visual, FPN and DMN.

**Regional brain fingerprint during AD**

Finally, we further explored the pattern of spatial reconfiguration as expressed by the ICC nodal strength of each brain region. Nodal strength was derived only from the edges that were significantly different from a permuted matrix randomly including subjects of the three groups (see "Methods"). We observed that regions contributing to the fingerprint (fingerprinting hubs) differed across groups and between cohorts (cf. Supplementary Fig. 4), confirming that functional connectivity patterns are unique. Nevertheless, there was considerable overlap in the edges with the highest fingerprint across the two cohorts (Fig. 5).

At last, we wanted to link brain fingerprints during the different stages of AD with behaviour. After deriving the group-specific nodes with the highest ICC values that were common across the two independent cohorts (cf. "Methods"), we applied a Neurosynth meta-analysis based on 50 topic terms onto the brain fingerprint of each group, similarly to previous work[61,62]. We found that in CU Aβ−, brain fingerprints were associated with higher-order processes such as long-term memory, language, semantics and executive functions, that rely on the integration of complex mental representations. On the other hand, in MCI Aβ+ we observed a shift towards

low-order sensory and motor processes. This pattern aligns with the 'cascading network failure' model, which posits a transition in connectivity from higher-order to lower-order circuits. This shift may represent an initial compensatory mechanism in the early stages of the disease, which eventually falters as the disease progresses[63].

In AD dementia we observed a shift back to high-order cognitive functions, similar to CU Aβ−, along with the involvement of regions related to affective/social processing including 'social cognition', 'pain', 'emotion' and 'reward-based decision making'.

## Discussion

In this work, we aimed to provide an answer to the fundamental question of how human brain fingerprints change during normal and pathological brain ageing due to AD. To do so, we evaluated the within-session identification properties of FCs during healthy ageing (CU Aβ−), MCI and dementia due to AD (i.e. MCI Aβ+ and AD dementia) in two independent cohorts using fMRI data from $N = 126$ individuals, and along two main lines: i) we investigated whether the within-session identification properties were maintained at the whole-brain level across the different clinical groups, ii) we explored the spatial configuration of brain fingerprints along the continuum between healthy and pathological brain ageing due to AD (Fig. 1).

Our study revealed that individuals can be accurately identified based solely on the patterns of brain activity even during AD (Fig. 2). Remarkably, individuals remained highly consistent in their brain connectivity across test and retest (ISelf), independent of their clinical condition. Additionally, patients exhibited a high degree of distinguishability from one another (IOthers), sometimes even more so than healthy elderly individuals among themselves (Supplementary Fig. 1B). This finding suggests that disease can enhance the distinctiveness of individuals based on their functional connectivity. We note that these results were observed in two independent datasets (Geneva and ADNI) with different acquisition parameters, signal-to-noise ratio and pre-processing pipelines, supporting the robustness of our results. This highlights the importance of taking into consideration, and making use of, this rich inter-individual variability to fully capture the complexity of functional alterations associated with AD. For example, tau spread has been linked to averaged templates of functional connectivity[50] and our findings carry important implications for this field of research, as they suggest that leveraging the individual characteristics of FCs could be fundamental to elucidating the heterogeneity of tau spread and, therefore, disease progression.

When examining the topological distribution of connections underlying this uniqueness, we observed a spatial reconfiguration of regions with the highest fingerprint during different stages of AD. Consistent with previous research utilizing MEG-based FCs[16], we identified a global decrease in identification between healthy elderly individuals and those with MCI. Additionally, and extending this work, we reported a significant increase in the number and sparsity of these temporally stable connections in patients with AD dementia (Figs. 3C and 4A). In MCI Aβ+, the AD pathology is at its early stages, and it is conceivable that this can lead to functional connectivity reconfiguration and readaptation, resulting in a reduced number of connections with stable patterns of connectivity among individuals. In our study, we included only MCI Aβ+, which is a biologically homogeneous subtype with a relatively predictable clinical trajectory[57,58]. However, not all MCI Aβ+ convert to AD dementia and MCI Aβ+ can present with distinct clinical subtypes (e.g. amnestic vs non-amnestic). Thus, we cannot exclude that the reduced number of regions contributing to the fingerprint may also reflect this clinical heterogeneity. Conversely, AD dementia represents a more well-defined clinicopathological entity[51,64], and the advanced stage of the disease facilitates, in most cases, clinical exclusion of alternative aetiologies, leading to a higher clinical homogeneity between patients. In addition, the AD pathophysiology is advanced in these patients and the functional reconfiguration is likely to have reached its plateau, and this could result in a higher number of connections remaining "unhealthily" stable across time among individuals.

Next, we observed that in CU Aβ−, in the visual (VIS), fronto-parietal (FPN), default-mode (DMN) and limbic (L) networks, edges with the highest fingerprint were mostly within-networks, while they appeared increasingly in between-networks connections in AD dementia (Figs. 3 and 4B). Previous work consistently showed that the FPN and DMN networks are found among the networks that display the highest inter-subjects variability in healthy populations[9,65,66] and our results showed that the variability initially intrinsic to within-system connections gets distributed to connections between these networks and the rest of the brain, as the disease progresses. Notably, our findings in AD dementia patients showed that having increasing temporally stable and therefore differentiable links does not always carry a positive prognostic. One possible hypothesis would be that the neurodegenerative processes affect the healthy topological variability of functional connectivity patterns, creating unhealthy "hyperstability" in the functional connections between different functional systems, which then could hinder the capacity of the brain networks to hop[67] or reconfigure[68] between different dynamical states. This aligns with prior research demonstrating that variability in brain functions is crucial for ensuring the brain's optimal responsiveness to a dynamic environment, and that this characteristic diminishes with age[69–71] and generally supports cognitive performance[70,71].

Previous works showed that the main drivers of the uniqueness of each individual functional connectome originate from brain areas responsible for higher-order cognitive processing during health[9,22]. However, it was not known whether this changed in response to AD. In this work, we present strong evidence based on two independent cohorts, revealing how within-session brain fingerprints map onto different cognitive functions during healthy vs pathological ageing due to AD (Fig. 5). Specifically, during healthy ageing, brain fingerprints exhibit a range of associations with higher-order processes, resembling those observed in young, healthy individuals in between-sessions fingerprints. Conversely, in MCI Aβ+, brain fingerprints show a shift towards lower-order sensory-motor processing. In the early stages of the disease, as amyloid pathology accumulates and affects connectivity within regions responsible for higher-order cognition, such as the DMN[45], it is plausible that these perturbations may lead to decreased regularity and stability in their connectivity patterns over time, resulting in a less distinctive 'fingerprint'. In contrast, sensory-motor regions, typically unaffected by early amyloid pathology[72], may exhibit adaptive changes in their functional connectivity patterns as a compensatory mechanism, in line with the 'cascading network failure' model proposed by Jones et al.[63]. Finally, as AD progresses to its advanced stages, we have observed a shift in fingerprint towards higher-order cognitive functions, encompassing social, affective, emotional and pain processing as well as decision making influenced by reward. The involvement of regions modulating emotion and pain is consistent with clinical and imaging observations indicating that social-emotional functioning tends to remain relatively preserved in AD. This preservation aligns with hyperactivity observed in the SAL network, known for its role in detecting and integrating emotional and sensory stimuli[73]. In the context of the cascading network failure model, these circuits may represent a final opportunity for the brain to 'compensate' for AD pathology. On the contrary, the highest fingerprints in regions implicated in higher-order processes, may denote maladaptive stability in regions where functional reconfiguration is no more possible. Note that brain fingerprints were linked with memory processes both in healthy ageing and during AD dementia (cf. Fig. 5B), although the regions driving this association differed in the 3 groups (cf. Fig. 5A).

This study builds upon an existing body of research in AD utilizing various neuroimaging modalities and analytical techniques, including machine learning approaches, giving insights into individuality. The contribution of this work lies in the novel approach we have taken to individual feature extraction within each specific diagnostic group. While machine learning approaches extract features from existing data and diagnostic classifications that allow new patient allocation to a specific diagnostic group, this approach extracts sources of individual variability specific to a diagnostic group that could help in predicting individual clinical features within each diagnostic group. In this respect, the extraction of individualized

FC features through our fingerprinting approach could potentially be used for machine learning prediction. The two approaches are therefore complementary and answer different clinical and research questions.

Our work has some limitations. First, the impact of the choice of the brain atlas should be further verified. Second, it is known that connectivity measures are highly susceptible to artefacts arising from head motion and respiratory fluctuations[74,75], and these effects are even more pronounced in pathological conditions. However, in our datasets, we did not find high-motion data points to significantly contribute to the differences in fingerprinting across groups. We observed no significant difference across groups in the percentage of censored volumes [$p \geq 0.634$] and differences in motion between test and retest volumes did not explain the variance of individuals' test–retest similarity (i.e. ISelf, cf. Supplementary Table 1A). Nonetheless, future work should analyse in-depth the effect of motion at shorter time scales, where these artefacts can dominate. Third, in this study, we used two halves of the same scanning session to estimate identifiability, focusing more on the FC features leading to brain identification within the session. Obtaining test–retest sessions across different days in clinical cohorts poses significant challenges, and to the best of our knowledge, there are currently no publicly accessible large datasets of fMRI data collected across closely spaced time points (such as one week apart) in cognitively impaired cohorts, unlike with healthy cohorts (e.g. Human Connectome Project). One potential workaround to this issue is to cut the resting state time series in half, as originally proposed[22]. Although this approach introduces the confound of looking at "within-session" fingerprinting, which could be influenced by the specific moment of scanning, it has the advantage of reducing scanner and acquisition noise, which are typically major confounding factors in connectome identification[23]. Moreover, this method has been shown to yield similar identifiability results compared to data acquired across separate sessions (refer to Supplementary Fig. 3 in ref. 22 for a comparison of healthy subjects' data from the Human Connectome Project). Nonetheless, it should be noted that within-scan fingerprinting in this work should be regarded more as a first temporal stability investigation of the resting-state functional brain network across AD. Future studies should aim to replicate our analysis using the "standard" between-sessions identification approach.

The scope of the current work was to investigate the preservation of identifiability and characterize its topological configuration across the different stages of cognitive decline due to AD. This study answers this fundamental question by pioneering the use of fMRI data from amyloid-β positive Alzheimer's patients across various cognitive decline stages, including dementia, and across two independent cohorts, marking a significant advancement in the field. While the ultimate goal is to utilize individual features of FC to predict behavioural and clinical outcomes, such as cognitive scores, this was not possible in this current work for the following reasons. First, individual variations in clinical features may not be fully characterized by global assessments like the MMSE, which was the only behavioural score consistently available across all subjects in the two datasets. Additionally, MMSE is used as a criterion for patient stratification (e.g. CU: MMSE ≥ 27; MCI: MMSE ≥ 19), therefore attempting to predict MMSE scores from FC individual variability within specific patient groups risks circular reasoning and may be confounded by the limited variance observed in MMSE scores within each group (e.g. in ADNI: CU Aβ− M(SD) = 29.2(0.9); MCI Aβ+ M(SD) = 27.2(2.4); AD dementia M(SD) = 21.6(2.7)), thereby compromising statistical power. Future work should focus on predicting clinical scores utilizing more sensitive neuropsychological assessments beyond MMSE.

This work raises novel questions and indicates directions for upcoming research. For example, future works could build upon these findings by examining the temporal aspects of brain fingerprints in neurodegeneration. Recent studies have in fact demonstrated that, in healthy individuals, brain fingerprints emerge at different time-scales[76], yet it is unknown whether this could change because of AD. Future studies should integrate and validate our findings over longitudinal data to directly link temporal stability changes to adaptive or maladaptive brain function in neurodegeneration. Subsequent studies should also delve deeper into the relationship between

atrophy, tau and amyloid accumulations. In our study, both cohorts were stratified based on amyloid status and stages of cognitive decline. Future investigations with larger samples could explore differences in groups with stratification using comprehensive biomarker phenotyping, such as the presence of tau pathology, atrophy, and/or APOE genotyping.

## Methods

### Participants

Participants were included from two independent cohorts: the Geneva Memory Centre (Geneva University Hospitals, Geneva, Switzerland) and the Alzheimer's disease Neuroimaging Initiative (ADNI). The Geneva cohort, from a clinical setting, offers insights into real-world AD presentation, although its characterization may be shaped by the healthcare management of this particular centre. The multicentric ADNI cohort, commonly used for validation, provides diverse geographic representation, enhancing the generalizability and reproducibility of the results. The demographic consistency (cf. 'Statistics and reproducibility' section, inclusion criteria and clinical assessment in 'Clinical assessment' section) across both cohorts contribute to a more robust extrapolation of findings to the wider AD population.

Inclusion criteria were availability of (i) fMRI and T1-weighted scans, (ii) 18F-Florbetapir or 18F-Florbetaben amyloid-PET to derive amyloid β-status (iii) clinical and cognitive assessments and demographic data, and (iv) identical fMRI acquisition parameters (cf. 'Image acquisition parameters' sections). The exclusion criterion was the presence of any significant neurologic disease other than AD (cf. 'Clinical assessment' section). Subjects ranged from healthy ageing and CU Aβ−, to MCI Aβ+, and Aβ+ subjects with dementia, i.e., AD dementia (cf. 'Clinical assessment' section for details about clinical and Aβ-status). Demographics are provided in the 'Statistics and Reproducibility' section.

### Clinical assessment

Clinical status was established by expert neurologists of the Geneva Memory Centre (cf.[77] for full details on the clinical assessment) for the Geneva cohort, and from ADNI collaborators for the ADNI cohort (cf. https://adni.loni.usc.edu/wp-content/themes/freshnews-dev-v2/documents/clinical/ADNI3_Protocol.pdf for full details on the clinical assessment). In brief, CU were individuals with or without subjective cognitive complaints and an absence of significant impairment in cognitive functions or activities of daily living (Geneva cohort: MMSE ≥ 27, non-depressed; ADNI: MMSE ≥ 27 and clinical dementia rating (CDR) = 0, non-depressed, cf. Supplementary Table 2). MCI were subjects with objective evidence of cognitive impairment, cognitive concern reported by the patient and/or informant (family or close friend), and little or no functional impairment in daily living activities (Geneva: MMSE ≥ 19) (ADNI: MMSE ≥ 19, CDR = 0.5). Individuals living with dementia were defined based on the same above criteria for MCI, but with impairment in the activities of daily living and fit the NINCDS/ADRDA criteria for probable AD (GENEVA: MMSE = 12-20) (ADNI: MMSE = 17–26, CDR ≥ 0.5).

For both datasets, the exclusion criterion was the presence of any other significant neurologic disease. These included: Parkinson's disease, multi-infarct dementia, Huntington's disease, normal pressure hydrocephalus, brain tumour, progressive supranuclear palsy, seizure disorder, subdural haematoma, multiple sclerosis, or history of significant head trauma followed by persistent neurologic deficits or known structural brain abnormalities (cf. https://adni.loni.usc.edu/wp-content/themes/freshnews-dev-v2/documents/clinical/ADNI3_Protocol.pdf).

### Amyloid-β status

In the Geneva cohort, amyloid-β deposition was measured using 18F-florbetapir or 18F-flutemetamol PET, using standard imaging protocol, reconstructions and pre-processing pipelines, previously described in detail[78]. Given the use of two different amyloid-PET tracers, the standardized uptake value ratio (SUVr) was converted to a common scale, the Centiloid (CL) scale, a standardisation method proposed to harmonise the

results obtained across tracers[79]. Aβ-status was determined in two ways: using the previously established cut-point (CL > 12[80]) and visually determined by an expert nuclear medicine physician (VG, >15 years of experience in the field) using visual assessment and standard operating procedures approved by the European Medicines Agency[81,82]. In two discordant cases, where the CL was borderline, the visual assessment (positive) was preferred.

In ADNI, Aβ-status was determined using global amyloid-PET SUVR, derived after whole cerebellum normalisation of [18]F-florbetapir or [18]F-florbetaben PET, following pre-established reconstruction and pre-processing protocols (cf. https://adni.loni.usc.edu/methods/pet-analysis-method/pet-analysis/) and cut-points (global AV45 SUVR > 1.11; global FBB SUVR > 1.08)[83]. As in the Geneva cohort, to allow aggregation of data from the two tracers, the global amyloid-PET SUVR values were converted to the Centiloid scale[79] and reported in Supplementary Table 2.

### Image acquisition parameters

**Geneva.** Structural and functional data were acquired using a 3 T Siemens Magnetom Skyra scanner (Siemens Healthineers, Erlangen, Germany) using a 64-channel phased-array head coil. Scans were performed within the radiology and neuroradiology division, at Geneva University Hospitals, Geneva, Switzerland. An EPI-BOLD sequence was used to collect functional data from 35 interleaved slices (slice thickness = 3 mm; multi-slice mode = interleaved; FoV = 192 × 192 × 105 mm; voxel size = 3 mm isotropic; TR = 2000ms, TE = 30 ms; flip-angle = 90°; GRAPPA acceleration factor = 2, time points = 200, approximate acquisition time = 7 min). Whole-brain T1-weighted anatomical images were acquired using a 3D MPRAGE sequence (slice thickness = 0.9 mm; FoV =2 63 × 350 × 350 mm; voxel size = 1 mm isotropic; TR = 1930 ms; TE = 2.36 ms, flip-angle = 8°; GRAPPA acceleration factor = 3).

**ADNI.** For ADNI, data was obtained using 3 T MRI scanners with a standardised protocol across imaging sites (full details in https://adni.loni.usc.edu/wp-content/uploads/2017/07/ADNI3-MRI-protocols.pdf). An EPI-BOLD sequence was used to acquire functional data (slice thickness = 3.4 mm, FoV=220 × 220 × 163 mm, voxel size = 3.4 isotropic; TR = 3000 ms; TE = 30 ms; flip angle = 90°; GRAPPA acceleration factor = 2; time points = 197, approximate acquisition time = 10 min). Whole-brain T1-weighted anatomical images were acquired with a 3D MPRAGE sequence (slice thickness = 1 mm, FoV = 208 × 240 × 256 mm; voxel size = 1 × 1 × 1 mm; TR = 2300 ms, TE = 3 ms, flip angle = 9°, GRAPPA acceleration factor = 3).

### Image processing

Image processing pipelines for the two cohorts included substantially similar steps, yet with some small differences (details below). Results are therefore not only replicated across different cohorts, but also irrespective of minor differences in preprocessing choices.

**Geneva.** fMRI data were preprocessed using in-house MATLAB code based on state-of-the-art fMRI processing guidelines[52,75,84]. Below follows a brief description of these steps. Structural images were first denoised to improve the signal-to-noise ratio[85], bias-field corrected, and then segmented (FSL FAST) to extract white matter (WM), grey matter (GM) and cerebrospinal fluid (CSF) tissue masks. These masks were warped in each individual subject's functional space by means of subsequent linear and non-linear registrations (FSL flirt 6 dof, FSL flirt 12 dof and FSL fnirt). The following steps were then applied on the fMRI data: BOLD volume unwarping with applytopup, slice timing correction (slicetimer), realignment (mcflirt), normalisation to mode 1000, demeaning and linear detrending (MATLAB detrend), regression (MATLAB regress) of 18 signals: 3 translations, 3 rotations and 3 tissue-based regressors (mean signal of whole-brain, WM and CSF, as well as nine corresponding derivatives (backwards difference; MATLAB). We tagged high head-motion volumes on the basis of three metrics: frame displacement (FD, in mm), standardised DVARS[86] (D referring to temporal derivative of

BOLD time courses, VARS referring to root mean square variance over voxels) as proposed in ref. 52, and SD (standard deviation of the BOLD signal within brain voxels at every time-point). The FD, DVARS were obtained with fsl_motion_outliers and SD vectors with MATLAB. Volumes were motion-tagged when FD > 0.3 mm and standardised DVARS > 1.7 and SD > 75th percentile +1.5 of the interquartile range, as per FSL recommendation[87]. Subjects (N = 4) with more than 30% motion-tagged volumes were excluded from the analyses. For the remaining subjects, the tagged volumes were not removed. There was no significant difference across groups in the percentage of tagged volumes [p = 0.638] and FD [p = 0.920]. There was no difference across test and retest in the percentage of tagged volumes [p = 0.565] nor in the average FD [p = 0.693]. Note that motion was accounted for in our statistical analyses (see section "Functional connectivity and whole-brain within-session brain-fingerprint").

A bandpass first-order Butterworth filter [0.01 Hz and 0.15 Hz] was applied to all BOLD time-series at the voxel level (MATLAB butter and filtfilt). The choice of the bandpass filter was aligned with previous works[76] where the choice of the filtering proved to be meaningful to capture the effect of brain fingerprints in the temporal domain, and in the analogy between MEG and fMRI fingerprints[15], and with respect to the relationship between brain fingerprints and structure-function coupling[88].

The first three principal components of the BOLD signal in the WM and CSF tissue were regressed out of the GM signal (MATLAB, pca and regress) at the voxel level. A whole-brain data-driven functional parcellation based on 248 regions including cortical and subcortical areas as obtained by ref. 89, was projected into each subject's T1 space (FSL flirt 6 dof, FSL flirt 12 dof and finally FSL fnirt) and then into the native EPI space of each subject. We also applied FSL boundary-based-registration[90] to improve the registration of the structural masks and the parcellation to the functional volumes.

In some rare cases, BOLD signal from some ROIs was missing. When signal from an ROI was not available in more than 10% of subjects it was excluded from the analyses for all; this concerned a total of eight ROIs, corresponding to seven subcortical and one cortical ROIs. In the remaining few cases with no signal, metrics were computed with available data only.

**ADNI.** Anatomical and functional images were preprocessed with a standardised in-house-developed preprocessing pipeline[86] implemented in MATLAB (MATLAB 2021a version 9.10; MathWorks Inc., Natick, MA, USA) and including functions from SPM8 and SPM12 (http://www.fil.io-n.ucl.ac.uk/spm/). Individual structural T1 images were registered to each individual's functional space (SPM coreg) while keeping the high T1 resolution, and segmented into GM, WM and cerebrospinal fluid (SPM new segment). Functional scans were realigned (SPM realign) and spatially smoothed (SPM smooth) with a Gaussian filter with FWHM = 5 mm. Nuisance signals were regressed out by means of a GLM, specifically linear and quadratic trends, 6 motion parameters and average signals in the WM and cerebrospinal fluid. The same whole-brain data-driven functional parcellation[89] used for the Geneva dataset was adopted here to extract regional timecourses. The parcellation in MNI coordinates was first normalised to the individuals' previously registered T1 images (functional space, structural high resolution), and then resampled to the lower functional BOLD resolution. Regional time series were then extracted by averaging the voxelwise pre-processed BOLD signals within each of the 248 regions of the parcellation. Finally, regional timecourses were band-pass filtered with cut-offs of 0.01–0.15 Hz to isolate typical resting-state fluctuations. Refer above for discussion about the choice of the bandpass filter.

The FD vectors (obtained from SPM head motion parameters using the procedure described in ref. 75) were used to tag outlier BOLD volumes with FD > 0.5 mm as per recommendation in ref. 76. Subjects (N = 7) with more than 30% motion-tagged volumes were excluded from the analyses. For the remaining subjects, the tagged volumes were not removed. There was no significant difference across groups in the average FD [p = 0.061] but there was a significant difference for tagged volumes [p = 0.048], revealing that the number of tagged volumes, but not FD, was higher in AD dementia

and MCI Aβ+. There was no difference across test and retest in the percentage of tagged volumes [$p = 0.065$], but there was a significant difference in the average FD [$p < 0.001$], revealing that FD was higher in the second part of the acquisition. To factor out the effect of motion in the brain fingerprint, motion was added as a nuisance variable in the whole brain fingerprinting analyses (cf. section 'Whole-brain within-sessions brain fingerprint during AD').

In some rare cases, the BOLD signal from some ROIs was missing. When the signal from an ROI was not available in more than 10% of subjects it was excluded from the analyses for all; this concerned a total of two ROIs in subcortical regions. In the remaining few cases with no signal, metrics were computed with available data only.

### Functional connectivity and whole-brain within-session brain-fingerprint

We estimated individual FC matrices using Pearson's correlation coefficient between the averaged signals of all region pairs. The resulting individual FC matrices were composed of 248 nodes, as obtained by ref. 89. Finally, the resulting FCs were ordered according to seven cortical resting state networks (RSNs) as proposed by ref. 91, plus one additional network including subcortical regions (similarly to ref. 92, see also Fig. 1A).

We estimated within-session identifiability or fingerprinting, following the approach proposed in ref. 22. This method involves splitting the fMRI times series in halves and enables quantification of within-session connectome fingerprints. Previous work has demonstrated that this method produces very similar results to those obtained from data acquired across separate sessions (i.e. between-sessions fingerprint) in healthy subjects from the Human Connectome Project (HPC) (see Supplementary Fig. 3 in ref. 22). Although within and between-sessions fingerprinting held similar results, they are different approaches to quantifying the brain-fingerprint, and this should be compared in future studies. However, we note that there are currently no clinical datasets available that include two fMRI sessions acquired within a short-time gap (i.e. within around one or two weeks). Therefore, within-session fingerprint is currently the only method available for estimating brain-fingerprint during AD. In this current study, we estimated identifiability across the first half (test) and second half volumes (retest) within the same scan. Recent work has shown that a good level of identifiability across the different RSNs can be reached from around 200 s (see Fig. 4B, in ref. 76). In this work, each test and retest session had 100 volumes (Geneva) or 90 volumes (ADNI) with a TR of 2 s and 3 s, respectively, therefore providing sufficient data for achieving a good success rate and identifiability across the entire brain.

At the whole-brain level, the fingerprint was calculated for each subject $s$ as test–retest similarity between FCs (cf. Fig. 1B; we called this metric *ISelf*).

$$\text{ISelf}(s) = \text{corr}\big(FC_{test}(s), FC_{retest}(s)\big)$$

Then, for each subject $s$ we computed an index of the FCs similarity with the other subjects $i$ in their group (*IOthers*), where $N$ is the total number of subjects in each group:

$$IOther(s) = \frac{\sum_{i \neq s} \big(corr\big(FC_{test}(s), FC_{retest}(i)\big) + corr\big(FC_{retest}(s), FC_{test}(i)\big)\big)}{2N - 2}$$

A second metric, *IDiff* (Fig. 1B), provides a group-level estimate of the within- (*ISelf*) and between-subjects (*IOthers*) test–retest reliability distance, where *Subj* is the set of subjects:

$$IDiff = \underset{s \in \text{Subj}}{\text{mean }} ISelf(s) - \underset{s \in \text{Subj}}{\text{mean }} IOthers(s)$$

Finally, we measured the success rate[9] of the identification procedure as the percentage of cases with higher within- (*ISelf*) vs between-subjects (*IOthers*) test–retest reliability. These metrics have been introduced and estimated in healthy populations in previous work[22].

### Spatial specificity of brain fingerprint: edge-wise intra-class correlation.

Spatial specificity of FC fingerprints was derived using edgewise intra-class correlation (ICC) with a one-way random effect model according to[93] (cf. Fig. 1C):

$$ICC = \frac{MS_R - MS_W}{MS_R + (k-1)MS_W}$$

where $MS_R$ = mean square for rows (between the subjects); $MS_W$ = mean square for residual sources of variance; $k$ = sessions. ICC coefficients quantify the degree of similarity between observations/measures and find high applicability in reliability studies[94]. The higher the ICC coefficient, the stronger the agreement between the two observations[22,76], to quantify the similarity between test and retest for each edge (FC between two regions). A high ICC indicates that a larger proportion of the variance across test and retest is due to differences between the subjects, rather than differences between test and retest or random error. A low ICC, on the other hand, indicates that there is more variability due to differences between test and retest or random error, than due to differences between subjects. In other words, the higher the ICC of an edge, the more that edge connectivity is similar for each subject across test and retest, as well as the variability across subjects, i.e. the higher the 'fingerprint' of that edge.

The ICC has been commonly used in the functional connectivity literature to assess reliability ([22,74,95–97]). A few strengths of the ICC include: (1) the ability to assess absolute agreement in repeated measurements of an object (unlike, for example, Pearson's correlation, wherein variables are scaled and cantered separately), (2) the ability to explicitly model multiple known sources of variability (e.g. scanner, brain response, head motion) and (3) comparing within and between variability across the objects of measurement. Depending on whether and how sources of error (or "facets"; e.g. scanner) may be specified, one of three ICC forms may be used[98]. In brief, usage is as follows: ICC (1,1) is used to estimate agreement in exact values when sources of error are unspecified; ICC (2,1) is often referred to as "absolute agreement" and is used to estimate agreement in exact values when sources of error are known (e.g. repeated runs) and modelled as random; and ICC (3,1) is often referred to as or "consistent agreement" and is used to estimate agreement in rankings when sources of error are known and modelled as fixed (resulting in a mixed effects ANOVA). In this paper we used ICC (1,1), following previous works[22], because variability in fMRI data can come from different known (e.g. scanner, head motion) and unknown sources. In the statistics literature the ICC is akin to a measure of discriminability and is commonly categorized as follows: poor < 0.4, fair 0.4–0.59, good 0.6–0.74 and excellent ≥ 0.75[99]. In this paper, this categorization was taken as reference for the thresholds on the ICC matrices, which was set at 0.6 (good[99]).

Edge-wise ICC was computed for all possible edges and for each group separately, with the aim to quantify the edges-wise functional connectivity fingerprint, distinctive of each clinical group. In order to control for sample size differences across groups, bootstrapping was used to accurately estimate edgewise fingerprints: for each group, ICC was calculated across test and retest for subsets of randomly chosen $N = 10$ (ADNI) or $N = 5$ (Geneva) subjects, across 1000 bootstrap runs, and then averaged within each group (Fig. 3A, B). Bootstrapping was performed in MATLAB using an in-house function. Matrices in Fig. 3A, B were binarized for ICC > 0.6 which is considered the lower threshold for a 'good' ICC score[55], and overlapping edges across the two cohorts were displayed in the binary ICC matrices (cf. overlap ICC matrix in Fig. 3C).

### Brain fingerprint in resting-state functional networks during AD

Next, we aimed to identify the commonalities in the distribution of edges with the highest ICC across different cohorts, both within and between resting-state networks. To achieve this, we analysed the overlap ICC binary matrix (Fig. 3C) and computed the following metrics. For each of one seven RSN, both within and between-networks (*net*), we quantified the number of overlapping ICC edges (ICC$_o$). Next, we computed a proportion of the ICC$_o$ edges relative to the total number of edges in each network and defined

it as $P_{ICC_o}$. Finally, and in order to determine the distance from the healthy reference (i.e. CU Aβ-), we computed the ratio over healthy individuals (Fig. 4A). The ratio ($R$) was calculated as follow, where $D$ = disease (i.e. MCI Aβ+ and AD dementia) and $H$ = healthy (i.e. CU).

$$R(net) = \frac{P_{ICC_o}D}{P_{ICC_o}H} - 1$$

### Between-groups significance of nodal brain fingerprint

In these analyses, we aimed to identify regions (or nodes) whose functional connectivity with the rest of the brain could account for significant differences in subject variability across the three groups. First, to isolate edges that were significantly different across groups, we compared real ICC matrices with a surrogate one. The surrogate ICC matrix was obtained from FCs of five randomly selected subjects from each group, the procedure was repeated for 1000 permutation runs and then averaged across runs to generate a surrogate group-unspecific ICC matrix for each cohort (Supplementary Fig. 3A). This permutation was implemented in Matlab R2022a using in-house code. Next, a $p$-value was computed for each edge as a proportion of times across permutation runs where surrogate values were bigger than the real value; edges with $p < 0.05$ were considered significant (see Supplementary Fig. 3B). Only significant edges were selected and a new matrix including significant edges with its real ICC value and having zeros for all the non-significant ones was used to compute nodal strength ICC. Nodal strength was computed as average, including zeros to account for non-significant edges, and rendered on the cortical surface using BrainNet[100] (cf. Supplementary Fig. 4). Next, to select the group-specific nodes with the highest ICC values that were common across the two independent cohorts, binary masks were obtained by selecting the top 25 percentile of ICC nodal strength and the overlap between the two cohorts was displayed (Fig. 5A). Each binary mask obtained that way provides a nodal representation of the brain region "hubs" involved in FC fingerprints in each group specifically.

### Brain fingerprint across AD and behaviour

A Neurosynth meta-analysis (https://neurosynth.org/), similar to the one implemented in previous studies[61,62], was conducted to assess cognitive functions associated with brain fingerprints at the different stages of AD. The procedure outputs, for each combination of brain fingerprint mask and cognitive function binary mask, a nodal z statistic that quantifies the similarity between the two maps. For brain fingerprint, we used the binary overlap masks in Fig. 5A—i.e. those including the ICC hubs with the highest fingerprint across the two cohorts. For cognition, we used the brain binary maps related to 50 topic terms common in the neuroimaging literature[74,101] derived from the Neurosynth database. These fingerprint and cognition maps were used as input for the meta-analysis to find significant associations between the ICC hubs or fingerprint masks and the brain cognitive functions Neurosynth maps. Last, we ordered the terms from high-order cognitive processing to sensory-motor processes similar to[62] for visualization, considering significant any association between group fingerprints and cognitive maps above $z > 3.1$[61,62] (Fig. 5B).

### Statistics and reproducibility

The statistical tests reported in this manuscript are two-sided and performed in RStudio 2022.07.2[102] and MATLAB R2022a[103]. Normality assumptions were checked prior to the analyses using the Shapiro–Wilk normality test (shapiro_test R function, rstatix package). When normality was not met, a non-parametric equivalent was used, as detailed below.

### Demographics Geneva. 

$N = 58$ subjects from the Geneva Memory Centre were included. Four subjects were excluded when motion-tagged volumes (see "Image processing" section) were >30%, leaving a total of $N = 54$ remaining subjects for analyses. These included $N = 16$ CU Aβ−, $N = 32$ MCI CU Aβ+ and $N = 6$ patients with AD dementia (cf. Supplementary Table 2 for all study-relevant covariates). Differences across groups in age, YoE and MMSE were tested using one-way ANOVA or its

non-parametric equivalent, i.e. the Kruskal–Wallis test; the Chi-square test was used for sex. Analyses were performed using RStudio. There were no differences in age [$p = 0.274$] and sex [$p = 0.363$] across groups, while AD dementia and MCI Aβ+ were on average significantly less educated than healthy individuals [$p = 0.004$]. As expected, MMSE [$p < 0.001$] and Centiloid [$p < 0.001$] scores varied across groups [$p < 0.001$], revealing lower cognition and higher amyloid load in MCI Aβ+ and AD dementia patients relative to CU Aβ−.

### Demographics ADNI. 

Data were obtained from the ADNI database (adni.loni.usc.edu). The ADNI was launched in 2003 as a public-private partnership, led by Principal Investigator Michael W. Weiner, MD. The primary goal of ADNI has been to test whether serial magnetic resonance imaging (MRI), positron emission tomography (PET), other biological markers, and clinical and neuropsychological assessment can be combined to measure the progression of MCI and early AD. $N = 79$ subjects from the ADNI database were included. Seven subjects were excluded when motion-tagged volumes (see "Image processing" section) were >30%, leaving a total of $N = 72$ subjects for analyses.

These were $N = 40$ CU Aβ−, $N = 21$ MCI Aβ+ and $N = 11$ AD dementia patients (cf. Supplementary Table 2 for all study-relevant covariates). Differences across groups in age, YoE and MMSE were tested using one-way ANOVA or its non-parametric equivalent, i.e. Kruskal–Wallis test; Chi-square test was used for sex. Analyses were performed using RStudio. There were no differences in age [$p = 0.848$] and sex [$p = 0.654$] across groups, while AD dementia were on average significantly less educated than healthy individuals [$p = 0.045$]. As expected, MMSE [$p < 0.001$] and Centiloid [$p < 0.001$] scores varied across groups [$p < 0.001$], revealing lower cognition and higher amyloid load in MCI Aβ+ and AD dementia patients relative to CU Aβ−.

Participants' sex was assigned at birth and was not based on self-report. Sex-based analyses were not conducted due to the insufficient sample size for meaningful subgroup analysis. There were no significant differences in the number of females and males within each group, so findings should be considered applicable to both sexes. While these results suggest generalizability, further research with a larger sample size would be beneficial to confirm these findings across both sexes.

### ISelf and IOthers. 

ISelf and IOthers were normally distributed in each group/cohort, with the exception of ISelf in CU Aβ− in the ADNI cohort. Analyses were performed using RStudio. To compare *ISelf* vs *IOthers* in each group/cohort we performed a paired-sample $t$-test or its non-parametric equivalent, i.e. Wilcoxon signed-rank test. Then, we used one-way ANOVAs to test the effect of the group on *ISelf* and *IOthers* separately after checking for nuisance variables, with 5000 permutations to control for sample size differences (using aovperm from permuco R-package) and to account for the normality violation[104]. For *ISelf*, the nuisance variables were age, sex, YoE, and the difference in motion between the test and retest scans, as the absolute difference between FD (*ISelf* ~ Group + Age + Sex + YoE + delta FD). For *IOthers*, the nuisance variables were also age, sex, and YoE, but motion (FD) was considered across the entire acquisition, as *IOthers* is a composite measure across test and retest (*IOthers* ~ Group + Age + Sex + YoE + FD). Additionally, when the scanner type varied across subjects (i.e. in ADNI), scanner type was also included as a nuisance variable for *IOther*. Finally, we did a permutation testing analysis to compare the *Success-rate* and *IDiff* from 1000 surrogate datasets of random ID matrices against the real value[15]. Permutation analyses were implemented in MATLAB using an in-house function where a permuted version of the ID matrix was built by randomly shuffling its elements.

### Within- vs between-functional networks edges. 

We aimed to determine whether there was a significant difference in the distribution of overlapping edges in within-networks vs between-networks for each group. To achieve this, we conducted a Chi-square test for each network comparing the number of edges in the within- vs the between-functional

networks. Significance was Bonferroni corrected for multiple comparisons: uncorrected *p*-value was multiplied for the number of comparisons, i.e. 8 = 7 RSN + subcortical regions (Fig. 4B). Analyses were performed using MATLAB.

## Reporting summary

Further information on research design is available in the Nature Portfolio Reporting Summary linked to this article.

## Inclusion and ethics statement

All subjects gave their informed consent for inclusion before they participated in the study, in accordance with the guidelines of the Declaration of Helsinki. All ethical regulations relevant to human research participants were followed.

The study protocol for the Geneva dataset was approved by Commission Cantonale d'Ethique de la Recherche sur l'être humain of the Geneva Canton (Switzerland).

ADNI obtained all IRB approvals and met all ethical standards in the collection of data. The following are the ethics committees and IRB boards that provided approval. The Ethics Committees/Institutional review boards that approved the ADNI study are: Albany Medical Center Committee on Research Involving Human Subjects Institutional Review Board, Boston University Medical Campus and Boston Medical Center Institutional Review Board, Butler Hospital Institutional Review Board, Cleveland Clinic Institutional Review Board, Columbia University Medical Center Institutional Review Board, Duke University Health System Institutional Review Board, Emory Institutional Review Board, Georgetown University Institutional Review Board, Health Sciences Institutional Review Board, Houston Methodist Institutional Review Board, Howard University Office of Regulatory Research Compliance, Icahn School of Medicine at Mount Sinai Program for the Protection of Human Subjects, Indiana University Institutional Review Board, Institutional Review Board of Baylor College of Medicine, Jewish General Hospital Research Ethics Board, Johns Hopkins Medicine Institutional Review Board, Lifespan—Rhode Island Hospital Institutional Review Board, Mayo Clinic Institutional Review Board, Mount Sinai Medical Center Institutional Review Board, Nathan Kline Institute for Psychiatric Research & Rockland Psychiatric Center Institutional Review Board, New York University Langone Medical Center School of Medicine Institutional Review Board, Northwestern University Institutional Review Board, Oregon Health and Science University Institutional Review Board, Partners Human Research Committee Research Ethics, Board Sunnybrook Health Sciences Centre, Roper St. Francis Healthcare Institutional Review Board, Rush University Medical Center Institutional Review Board, St. Joseph's Phoenix Institutional Review Board, Stanford Institutional Review Board, The Ohio State University Institutional Review Board, University Hospitals Cleveland Medical Center Institutional Review Board, University of Alabama Office of the IRB, University of British Columbia Research Ethics Board, University of California Davis Institutional Review Board Administration, University of California Los Angeles Office of the Human Research Protection Program, University of California San Diego Human Research Protections Program, University of California San Francisco Human Research Protection Program, University of Iowa Institutional Review Board, University of Kansas Medical Center Human Subjects Committee, University of Kentucky Medical Institutional Review Board, University of Michigan Medical School Institutional Review Board, University of Pennsylvania Institutional Review Board, University of Pittsburgh Institutional Review Board, University of Rochester Research Subjects Review Board, University of South Florida Institutional Review Board, University of Southern, California Institutional Review Board, UT Southwestern Institution Review Board, VA Long Beach Healthcare System Institutional Review Board, Vanderbilt University Medical Center Institutional Review Board, Wake Forest School of Medicine Institutional Review Board, Washington University School of Medicine Institutional Review Board, Western Institutional Review Board, Western University Health Sciences Research Ethics Board and Yale University Institutional Review Board.

## Data availability

The raw data that support the findings of this study for the ADNI dataset are available on the Alzheimer's Disease Neuroimaging Initiative (ADNI) platform (https://adni.loni.usc.edu/). Raw data generated at the Geneva site are available from the co-author Prof. Giovanni B. Frisoni, upon request. The derived FC matrices and behavioural data necessary to reproduce the main analyses of this study is available in Sara Stampacchia's GitHub repository (https://github.com/ss1913/fingerprints_alzheimer).

## Code availability

The full code necessary to reproduce the main results and figures is available in Sara Stampacchia's GitHub repository (https://github.com/ss1913/fingerprints_alzheimer).

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

## Acknowledgements

S.S. was supported by grants from the Swiss National Science Foundation [SNSF 320030_185028, PI: V.G.]. Data acquisition from the Geneva cohort was granted by SNSF 320030_185028 and 169876 (PI: V.G.) and Fonds Startup du département de radiologie et informatique médicale, Université de Genève, Faculté de Médecine (PI: S.S.). The Geneva Memory Centre (Centre de la Mémoire) is funded by the following private donors under the supervision of the Private Foundation of Geneva University Hospitals: A.P.R.A.—Association Suisse pour la Recherche sur la Maladie d'Alzheimer, Genève; Fondation Segré, Genève; Race Against Dementia Foundation, London, UK; Fondation Child Care, Genève; Fondation Edmond J. Safra, Genève; Fondation Minkoff, Genève; Fondazione Agusta, Lugano; McCall Macbain Foundation, Canada; Nicole et René Keller, Genève and Fondation AETAS, Genève. Competitive research projects have been funded by:

H2020 (project no. 667375), Innovative Medicines Initiative (IMI contract no. 115736 and 115952), IMI2, Swiss National Science Foundation (projects no. 320030_182772 and no. 320030_169876), VELUX Foundation. The Clinical Research Center, at Geneva University Hospital and Faculty of Medicine provides valuable support for regulatory submissions and data management. The authors thank Avid Radiopharmaceuticals Inc. for providing the 18F-Flortaucipir tracer without being involved in the data analysis or interpretation. E.A. acknowledges financial support from the SNSF Ambizione project "Fingerprinting the brain: Network science to extract features of cognition, behaviour and dysfunction" (grant number PZ00P2_185716). V.G. is supported also by the Velux Foundation, the Aetas Foundation, the Schmidheiny Foundation and research/teaching support through her institution from Siemens Healthineers, GE Healthcare, Novo Nordisk and Janssen. M.P. is partially supported by the Italian Ministry of Health (Ricerca Corrente). M.G.P. was supported by the CIBM Center for Biomedical Imaging, a Swiss research center of excellence founded and supported by Lausanne University Hospital (CHUV), University of Lausanne (UNIL), Ecole polytechnique fédérale de Lausanne (EPFL), University of Geneva (UNIGE) and Geneva University Hospitals (HUG). Data collection and sharing for the ADNI cohort was funded by ADNI (National Institutes of Health grant U01 AG024904) and DOD ADNI (Department of Defense award number W81XWH-12-2-0012). ADNI is funded by the National Institute on Aging, the National Institute of Biomedical Imaging and Bioengineering, and through generous contributions from the following: AbbVie, Alzheimer's Association; Alzheimer's Drug Discovery Foundation; Araclon Biotech; BioClinica, Inc.; Biogen; Bristol-Myers Squibb Company; CereSpir, Inc.; Cogstate; Eisai Inc.; Elan Pharmaceuticals, Inc.; Eli Lilly and Company; EuroImmun; F. Hoffmann-La Roche Ltd and its affiliated company Genentech, Inc.; Fujirebio; GE Healthcare; IXICO Ltd.; Janssen Alzheimer Immunotherapy Research & Development, LLC.; Johnson & Johnson Pharmaceutical Research & Development LLC.; Lumosity; Lundbeck; Merck & Co., Inc.; Meso Scale Diagnostics, LLC.; NeuroRx Research; Neurotrack Technologies; Novartis Pharmaceuticals Corporation; Pfizer Inc.; Piramal Imaging; Servier; Takeda Pharmaceutical Company; and Transition Therapeutics. The Canadian Institutes of Health Research is providing funds to support ADNI clinical sites in Canada. Private sector contributions are facilitated by the Foundation for the National Institutes of Health (www.fnih.org). The grantee organization is the Northern California Institute for Research and Education, and the study is coordinated by the Alzheimer's Therapeutic Research Institute at the University of Southern California. ADNI data are disseminated by the Laboratory for Neuro Imaging at the University of Southern California. We thank Benedetta Franceschiello, Emahnuel Troisi-Lopez, Pierpaolo Sorrentino, Valerio Zerbi and Jenya Chumin for the insightful comments. We are indebted to the patients, their carers, and the volunteers for taking part in this study, and to the clinical and research staff of the Geneva Memory Centre for data acquisition. We would like to acknowledge the entire ADNI leadership board and team for providing part of the data for this study. A complete listing of ADNI investigators can be found at: http://adni.loni.usc.edu/wp-content/uploads/how_to_apply/ADNI_Acknowledgement_List.pdf.

## Author contributions

Authors' contribution according to the CRediT taxonomy, see http://credit.niso.org/ for more information. Conceptualisation: S.S. and E.A. (lead). M.P., D.V.D.V., G.B.F., O.B. and V.G. contributed to the evolution of the overarching research goals and aims. Methodology: S.S. and E.A. (lead). D.V.D.V. contributed to the creation of analytical models. Software: S.S. and E.A. wrote the code for the analyses. Validation: S.S. Formal analyses: S.S. (lead), E.A., S.A., A.F. and M.G.P. contributed to fMRI data preprocessing. Resources: M.S., K.O.L., A.F., M.G.P., P.U., G.B.F., V.G. and E.A. Data curation: S.S. (lead), S.T., F.R., A.F., V.G. and M.G.P. contributed to data curation. Writing—original draft: S.S. (lead) and E.A. Writing—review and editing: S.S., E.A., V.G., O.B., D.V.D.V., M.S., F.R., M.G.P. and M.P. Visualisation: S.S. (lead) and E.A. Supervision: E.A. and V.G. Project administration: S.S., E.A. and V.G. Funding acquisition: S.S., E.A., V.G., O.B. and G.B.F.

## Competing interests

The authors declare the following competing interests: S.S. declares the award of a travel grant from the Organisation of Human Brain Mapping (OHBM) to present this work at their conference in June 2022. V.G. received research support and speaker fees through her institution from GE Healthcare, Siemens Healthineers, Novo Nordisk and Janssen.
