## [Peer Review File · Communications Biology]

Reviewers' comments:

Reviewer #1 (Remarks to the Author):

Major Comments:

Abstract: Clarity and Novelty

- The abstract provides an interesting overview of the study, however, it could benefit from a clearer articulation of the novelty and unique contributions of the work. Specifically, while it is stated that brain fingerprints change across health and disease, the direct implications of these findings for clinical practice, beyond a general call to consider individual variability, remain vague. A more explicit discussion of how these findings move the field forward or can be applied to improve patient outcomes would enhance the manuscript's impact.

Novelty and Contribution to the Field (General Comment):

The manuscript presents valuable insights into brain connectivity alterations in the context of cognitive decline and Alzheimer's disease (AD), leveraging functional MRI (fMRI) datasets from two independent cohorts. The concept of brain fingerprints, and their alterations across different stages of AD, is of high interest for the field of neuroimaging and neuroscience. However, the novelty aspect might be enhanced by a clearer delineation of how the presented findings advance beyond existing knowledge (e.g., uniqueness and variability of functional connectomes in cognitive decline) [Suggested Improvements: The authors should explicitly address the unique contribution their study makes against the backdrop of previously published literature. This includes a more detailed discussion highlighting specific novel findings about brain fingerprinting alterations in cognitive decline and AD that were not covered or were only partially addressed in prior studies.]

Methodological Specificity and Reproducibility (Multiple Sections):

Across the Methods section, there are several instances where the descriptions of the experiments and analyses lack sufficient specificity for replicability. For instance, the exact preprocessing steps for fMRI data (e.g., motion correction parameters, slices timing corrections) on Page 23 and onward, while mentioned, do not detail specific thresholds or criteria for inclusion/exclusion based on motion. Moreover, the description of permutation tests

ts and statistical analyses could benefit from more detail regarding the implementation (e.g., software used, exact statistical models).

fMRI Data Analysis (Page 24-25, "Image Processing"):

While the authors have endeavored to control for various confounding factors in their fMRI data analysis, certain technical details warrant further clarification. For example, the choice of bandpass filtering parameters [0.01 Hz, 0.15 Hz] and the impact of such filtering on the study outcomes need a more rigorous justification, as the filtering range can significantly influence FC analyses.

[Suggested Improvements: The authors should provide a justification for the choice of bandpass filtering parameters based on literature or preliminary analyses demonstrating the appropriateness of these parameters for capturing meaningful FC alterations. Additionally, considering the impact of motion artifacts, especially in the context of AD where patient cooperation might be challenging, further elaboration on motion correction methodologies and their efficacy is required.]

Statistical Analyses and Interpretation of Results (Page 27, "Functional Connectivity and whole-brain within-session brain-fingerprint"):

The statistical methods employed for identifying unique functional connectome fingerprints and their changes across different stages of cognitive decline and AD are indeed comprehensive. However, the manuscript lacks a detailed discussion on the limitations and assumptions of these methods, particularly the edgewise intra-class correlation (ICC) and its interpretation in the context of this study.

[Suggested Improvements: The authors should discuss the statistical assumptions underlying the use of ICC in their analysis, potential limitations, and how these might affect the interpretation of the results. Additionally, the authors might consider including sensitivity analyses to assess the robustness of their findings with respect to different statistical thresholds or parameters.]

Interpretation of Temporal Stability in MCI and AD Dementia:

The interpretation of increased temporal stability in AD dementia as potentially unfavorable ('hyperstability') and the decreased stability in MCI as potentially indicative of adaptive changes offers an innovative perspective. Nonetheless, this interpretation would be significantly strengthened by integrating or discussing longitudinal data or mechanistic studies that directly link temporal stability changes to adaptive or maladaptive brain function.

on in neurodegeneration. Without such data, these interpretations, while intriguing, remain somewhat speculative.

Minor Comments:

Consistency in Terminology (Throughout the manuscript):

The manuscript occasionally oscillates between terms such as "cognitive decline," "pathological aging," and "AD spectrum." Consistency in terminology will help maintain clarity and avoid potential confusion among readers. [Suggested Improvements: Standardize the terminology used throughout the manuscript to refer to AD and its stages, ensuring consistency and clarity in the narrative.]

Figure and Table References (General Comment):

There are several instances throughout the manuscript where figures and tables are referenced (e.g., Fig. 1, Table 1, Fig. S1); however, these are not included in the provided text, making it challenging to follow the data presentation and interpretation. [Suggested Improvements: Ensure that all referenced figures and tables are included in the manuscript, or provide a clear note if these are available in supplementary materials not provided for this review.]

Clarification on Cohorts and Sample Size (Page 20, "Participants and demographics"):

The manuscript details the inclusion of participants from two independent cohorts and provides demographic breakdowns. However, the rationale behind the cohort selection and its implications for the generalizability of the study findings could be elaborated further. [Suggested Improvements: Provide a brief discussion on why these particular cohorts were chosen and how the findings might or might not generalize to different populations considering the demographic composition of the included participants.]

Page 9, Overlap Between Cohorts in ICC Edges Analysis:

The description of the methods used to assess the overlap in ICC edges between cohorts could benefit from further clarity. Specifically, the criteria used to define 'good' ICC scores and to select top edges for inclusion in the analysis are mentioned but not thoroughly explained. Detailed criteria and justification for these methodological choices would enhance the reader's understanding.

Page 12-13, Chi-square Analysis of Network Edges

While the Chi-square analysis is a suitable method for comparing the proportions of edges within vs. between networks, the application of Bonferroni correction for multiple comparisons is mentioned but not detailed. The manuscript would benefit from a more explicit discussion of how many comparisons were made and the resulting significance threshold after correction.

Reviewer #2 (Remarks to the Author):

This is an interesting original study, conducted by experts in the field, and therefore with exceptional rigor, which uses connectivity-based brain fingerprinting concept to characterize AD.

I have a number of comments that hopefully will help improving the manuscript:

1. Brain fingerprint is a concept and should not be univocally associated with a specific neuroimaging technique. There are many pre-existing works on functional connectivity network-based person distinctiveness studies, notably with EEG, that would deserve to be cited to give a fairer context in the field.
2. There are already so many studies on connectivity network-based in AD from disparate structural and functional neuroimaging and electrophysiological data, not only fMRI. Correlation analysis, machine learning approaches, allow for example to give insights into individuality, just as the brain fingerprint approach. It's not clear in what, in essence, brain fingerprint brings something new compared to these existing approaches. Is there more accuracy? Is there better prediction? Is there new mechanisms/knowledge that we did not before? For example, in the abstract authors say "...by showing that functional connectivity profiles maintain their uniqueness, yet go through functional reconfiguration, during cognitive decline". This sentence is expected to be true for any brain network in which some links remain and other reconfigure across disease. Thus, to better appreciate the obtained findings the abstract, discussion and conclusion, should be shortened and reframed to improve clarity.
3. I'm not sure I found any correlation/prediction results between brain-fingerprint features and clinical scores (eg, MMSE). Without such analysis, and without its comparison with more standard brain features, it is impossible to evaluate the relevance of this approach for

personalized medicine, which is often advocated in the paper.

4. In conclusion, the overall length of the text, the resulting redundancy and the adopted phrasing makes difficult to appreciate the real value of the obtained results. I advise the authors to clarify the novelty of their results in a concise way and adopt a more coherent writing style

Response to reviewers

Reviewers' comments:

Reviewer #1 (Remarks to the Author):

Major Comments:

Major Comment 1

Abstract: Clarity and Novelty

- The abstract provides an interesting overview of the study, however, it could benefit from a clearer articulation of the novelty and unique contributions of the work. Specifically, while it is stated that brain fingerprints change across health and disease, the direct implications of these findings for clinical practice, beyond a general call to consider individual variability, remain vague. A more explicit discussion of how these findings move the field forward or can be applied to improve patient outcomes would enhance the manuscript's impact.

We thank the reviewer for this important remark. In line with the journal policies we have substantially reduced the length of the abstract (max 150 words) and in doing so we have provided a clearer and more concise articulation of this work and its implications.

Line 34:

Functional connectivity patterns in the human brain, like the friction ridges of a fingerprint, can uniquely identify individuals. Does this "brain fingerprint" remain distinct even during Alzheimer's disease (AD)? Using fMRI data from healthy and pathologically aging subjects, we found that individual functional connectivity profiles remain unique and highly heterogeneous during mild cognitive impairment and AD. However, the patterns that make individuals identifiable change with disease progression, revealing a reconfiguration of the brain fingerprint. Notably, connectivity shifts towards between-functional system connections in AD and to lower-order cognitive functions in early disease stages. These findings emphasize the importance of focusing on individual variability rather than group differences in AD studies. Individual functional connectomes could be instrumental in creating personalized models of AD progression, predicting disease course, and optimizing treatments, paving the way for personalized medicine in AD management.

Major Comment 2

Novelty and Contribution to the Field (General Comment):

The manuscript presents valuable insights into brain connectivity alterations in the context of cognitive decline and Alzheimer's disease (AD), leveraging functional MRI (fMRI) datasets from two independent cohorts. The concept of brain fingerprints, and their alterations across different stages of AD, is of high interest for the field of neuroimaging and neuroscience. However, the novelty aspect might be enhanced by a clearer delineation of how the presented findings advance beyond existing knowledge (e.g., uniqueness and variability of functional connectomes in cognitive decline) [Suggested Improvements: The authors should explicitly address the unique contribution their study makes against the backdrop of previously published literature. This includes a more detailed discussion highlighting specific novel findings about brain fingerprinting alterations in cognitive decline and AD that were not covered or were only partially addressed in prior studies.]

We thank the reviewer for pointing to this aspect. We have added the following sentences in the introduction.

Line 103:

This study builds upon prior research on brain fingerprinting in AD 1 by pioneering four significant advancements: a) employing fMRI for the first time, replacing the previously utilized MEG technology; b) broadening the scope of investigation to encompass the advanced stages of the disease, including dementia in addition to mild cognitive impairment (MCI); and c) exclusively enrolling amyloid-positive patients, thereby ensuring a focus on cognitive decline specifically attributable to AD and d) integrating data from two distinct cohorts, enhancing the robustness and generalizability of the findings.

Major Comment 3

Methodological Specificity and Reproducibility (Multiple Sections):

Across the Methods section, there are several instances where the descriptions of the experiments and analyses lack sufficient specificity for replicability. For instance, the exact preprocessing steps for fMRI data (e.g., motion correction parameters, slices timing corrections) on Page 23 and onward, while mentioned, do not detail specific thresholds or criteria for inclusion/exclusion based on motion.

We thank the reviewer for this comment and we changed the text as follows.

Line 547 (Geneva):

We tagged high head-motion volumes on the basis of three metrics: frame displacement (FD, in mm), standardised DVARS85 (D referring to temporal derivative of BOLD time courses, VARS referring to root mean square variance over voxels) as proposed in 42, and SD (standard deviation of the BOLD signal within brain voxels at every time-point). The FD, DVARS were obtained with fsl_motion_outliers and SD vectors with MATLAB. Volumes were motion-tagged when $FD > 0.3$ mm and standardised DVARS > 1.7 and $SD > 75$ th percentile $+1.5$ of the interquartile range, as per FSL recommendation 86. Subjects ($N=4$) with more than 30% motion-tagged volumes were excluded from the analyses. For the remaining subjects, the tagged volumes were not removed.

Line 594 (ADNI):

The FD vectors (obtained from SPM head motion parameters using the procedure described in 74) were used to tag outlier BOLD volumes with $FD > 0.5$ mm as per recommendation in 75. Subjects ($N=7$) with more than 30% motion-tagged volumes were excluded from the analyses (see section 1.0). For the remaining subjects, the tagged volumes were not removed.

Line 542 (Geneva):

The following steps were then applied on the fMRI data: BOLD volume unwarping with applytopup, slice timing correction (slicetimer, with slice order: interleaved), realignment (mcflirt), normalisation to mode 1000, demeaning and linear detrending (MATLAB detrend), regression (MATLAB regress) of 18 signals: 3 translations, 3 rotations, and 3 tissue-based regressors (mean signal of whole-brain, white matter (WM) and cerebrospinal fluid (CSF), as well as 9 corresponding derivatives (backwards difference; MATLAB).

Line 583 (ADNI):

Functional scans were realigned (SPM realign) and spatially smoothed (SPM smooth) with a Gaussian filter with $FWHM=5$ mm. Nuisance signals were regressed out by means of a GLM, specifically linear and quadratic trends, 6 motion parameters and average signals in the white matter and cerebrospinal fluid.

Major Comment 4

Moreover, the description of permutation tests and statistical analyses could benefit from more detail regarding the implementation (e.g., software used, exact statistical models).

We thank the reviewer for this comment and we provided details as required. Please note that to comply with the formatting guidelines, the statistics are now reported in the 'Statistics and Reproducibility' section.

Line 782:

Then, we used one-way ANOVAs to test the effect of group on ISelf and IOthers separately after checking for nuisance variables, with 5000 permutations to control for sample size differences (using aovperm from permuco R-package) and to account for the normality violation¹. For ISelf, the nuisance variables were age, sex, years of education (YoE), and the difference in motion between the test and retest scans, as absolute difference between FD (ISelf ~ Group + Age + Sex + YoE + delta FD). For IOthers, the nuisance variables were also age, sex, and YoE, but motion (FD) was considered across the entire acquisition, as IOthers is a composite measure across test and retest (IOthers ~ Group + Age + Sex + YoE + FD). Additionally, when the scanner type varied across subjects (i.e., in ADNI), scanner type was also included as a nuisance variable for IOther. Additionally, when the scanner type varied across subjects (i.e., in ADNI), scanner type was also included as a nuisance variable for IOther. Finally, we did a permutation testing analysis to compare Success-rate and IDiff from 1000 surrogate datasets of random ID matrices against the real value 46. Permutation analyses were implemented in MATLAB using in-house function where a permuted version of the ID matrix was built by randomly shuffling its elements.

Line 684:

In order to control for sample size differences across groups, bootstrapping was used to accurately estimate edgewise fingerprints: for each group, ICC was calculated across test and retest for subsets of randomly chosen N=10 subjects, across 1000 bootstrap runs, and then averaged within each group (Fig. 3A and 3B). Bootstrapping was performed in MATLAB using an in-house function.

Line 708:

First, to isolate edges that were significantly different across groups, we compared real ICC matrices with a surrogate one. The surrogate ICC matrix was obtained from FCs of five randomly selected subjects from each group, the procedure was repeated for 1000 permutation runs and then averaged across runs to generate a surrogate group-unspecific ICC matrix for each cohort (Supplementary Fig. 3A). This permutation was implemented in Matlab R2022a using in-house code. Next, a p-value was

computed for each edge as a proportion of times across permutation runs where surrogate values were bigger than the real value; edges with $p < .05$ were considered significant (see Supplementary Fig. 3B).

Major Comment 5

fMRI Data Analysis (Page 24-25, "Image Processing"):

While the authors have endeavored to control for various confounding factors in their fMRI data analysis, certain technical details warrant further clarification. For example, the choice of bandpass filtering parameters [0.01 Hz, 0.15 Hz] and the impact of such filtering on the study outcomes need a more rigorous justification, as the filtering range can significantly influence FC analyses. [Suggested Improvements: The authors should provide a justification for the choice of bandpass filtering parameters based on literature or preliminary analyses demonstrating the appropriateness of these parameters for capturing meaningful FC alterations. Additionally, considering the impact of motion artifacts, especially in the context of AD where patient cooperation might be challenging, further elaboration on motion correction methodologies and their efficacy is required.]

Thanks for this suggestion. We have now expanded on the choice of the bandpass in the manuscript, which now reads:

Line 562:

The choice of the bandpass filter was aligned with previous works (Van de Ville) where the choice of the filtering proved to be meaningful to capture the effect of brain fingerprints in the temporal domain, and in the analogy between MEG and fMRI fingerprints (Sareen et al.), and with respect to the relationship between brain fingerprints and structure-function coupling (Griffa et al.).

We acknowledge that motion artifacts can have an impact in the detection of brain fingerprints, particularly in clinical populations. We have carefully checked the impact of motion and we now expand on the findings in the Discussion section:

Line 407:

Second, it is known that connectivity measures are highly susceptible to artefacts arising from head motion and respiratory fluctuations 73,74, and these effects are even more pronounced in pathological conditions. However, in our datasets, we did not find high motion data points to significantly contribute to the differences in fingerprinting across groups. We observed no significant difference across groups in the percentage of motion-tagged volumes [$p \geq .634$] and differences in motion (FD) between test and

retest volumes did not explain the variance of individuals' test-retest similarity (i.e., ISelf, cf. Supplementary Table 1A). Nonetheless, future work should analyse in depth the effect of motion at shorter time scales, where these artefacts can dominate.

Major Comment 6

Statistical Analyses and Interpretation of Results (Page 27, "Functional Connectivity and whole-brain within-session brain-fingerprint"):

The statistical methods employed for identifying unique functional connectome fingerprints and their changes across different stages of cognitive decline and AD are indeed comprehensive. However, the manuscript lacks a detailed discussion on the limitations and assumptions of these methods, particularly the edgewise intra-class correlation (ICC) and its interpretation in the context of this study. [Suggested Improvements: The authors should discuss the statistical assumptions underlying the use of ICC in their analysis, potential limitations, and how these might affect the interpretation of the results. Additionally, the authors might consider including sensitivity analyses to assess the robustness of their findings with respect to different statistical thresholds or parameters.]

Thanks for this suggestion. We now expand on the ICC, as follows:

Line 666:

The ICC has been commonly used in the functional connectivity literature to assess reliability (e.g., Amico and Goni, 2018; Birn et al., 2013; Shah et al., 2016; Shehzad et al., 2009; Somandepalli et al., 2015). A few strengths of the ICC include: 1) the ability to assess absolute agreement in repeated measurements of an object (unlike, for example, Pearson's correlation, wherein variables are scaled and centered separately), 2) the ability to explicitly model multiple known sources of variability (e.g. scanner, brain response, head motion), and 3) comparing within and between variability across the objects of measurement. Depending on whether and how sources of error (or "facets"; e.g., scanner) may be specified, one of three ICC forms may be used (Noble et al., 2019). In brief, usage is as follows: ICC(1,1) is used to estimate agreement in exact values when sources of error are unspecified; ICC(2,1) is often referred to as "absolute agreement" and is used to estimate agreement in exact values when sources of error are known (e.g., repeated runs) and modeled as random; and ICC(3,1) is often referred to as or "consistent agreement" and is used to estimate agreement in rankings when sources of error are known and modeled as fixed (resulting in a mixed effects ANOVA). In this paper we used ICC(1,1), following previous works (Amico and Goni, 2018), because variability in fMRI data can come from different known (e.g., scanner, head

motion) and unknown sources. In the statistics literature the ICC is akin to a measure of discriminability and is commonly categorized as follows: poor <0.4, fair 0.4–0.59, good 0.6–0.74, excellent ≥ 0.75 (Cicchetti and Sparrow, 1981). In this paper, this categorization was taken as reference for the thresholds on the ICC matrices, which was set at 0.6 (good; Cicchetti and Sparrow, 1981).

Major Comment 7

Interpretation of Temporal Stability in MCI and AD Dementia:

The interpretation of increased temporal stability in AD dementia as potentially unfavorable ('hyperstability') and the decreased stability in MCI as potentially indicative of adaptive changes offers an innovative perspective. Nonetheless, this interpretation would be significantly strengthened by integrating or discussing longitudinal data or mechanistic studies that directly link temporal stability changes to adaptive or maladaptive brain function in neurodegeneration. Without such data, these interpretations, while intriguing, remain somewhat speculative

We agree with the reviewer. Unfortunately, longitudinal data are not available for this dataset at the moment (although the UK biobank is planning to scan longitudinal patients in the coming years). We are acknowledging this as a limitation in the Discussion section:

Line 449:

Future studies should integrate and validate our findings over longitudinal data to directly link temporal stability changes to adaptive or maladaptive brain function in neurodegeneration.

Minor Comments:

Minor Comment 1

Consistency in Terminology (Throughout the manuscript):

The manuscript occasionally oscillates between terms such as "cognitive decline," "pathological aging," and "AD spectrum." Consistency in terminology will help maintain clarity and avoid potential confusion among readers. [Suggested Improvements: Standardize the terminology used throughout the manuscript to refer to AD and its stages, ensuring consistency and clarity in the narrative.]

We thank the reviewer for this suggestion. We standardized the terminology throughout the manuscript to refer to it as AD and its stages.

Minor Comment 2

Figure and Table References (General Comment):

There are several instances throughout the manuscript where figures and tables are referenced (e.g., Fig. 1, Table 1, Fig. S1); however, these are not included in the provided text, making it challenging to follow the data presentation and interpretation. [Suggested Improvements: Ensure that all referenced figures and tables are included in the manuscript, or provide a clear note if these are available in supplementary materials not provided for this review.]

We thank the reviewer for this comment. All referenced main figures and tables are included in the manuscript and those reported in the Supplementary Materials are referred to as "Supplementary Fig. X). We regret that the Supplementary Materials were not provided for review and they are available in this new resubmission.

Minor Comment 3

Clarification on Cohorts and Sample Size (Page 20, "Participants and demographics"): The manuscript details the inclusion of participants from two independent cohorts and provides demographic breakdowns. However, the rationale behind the cohort selection and its implications for the generalizability of the study findings could be elaborated further. [Suggested Improvements: Provide a brief discussion on why these particular cohorts were chosen and how the findings might or might not generalize to different populations considering the demographic composition of the included participants.]

We thank the reviewer for this comment. We added the following text to the Participants section.

Line 460:

Participants were included from two independent cohorts: the Geneva Memory Centre (Geneva University Hospitals, Geneva, Switzerland) and the Alzheimer's Disease Neuroimaging Initiative (ADNI). The Geneva cohort, from a clinical setting, offers

insights into real-world AD presentation, although its characterization may be shaped by the healthcare management of this particular centre. The multicentric ADNI cohort, commonly used for validation, provides diverse geographic representation, enhancing generalizability and reproducibility of the results. The demographic consistency (cf. ‘Statistics and Reproducibility’ section), inclusion criteria (cf. below) and clinical assessment (cf. ‘Clinical assessment’ section) across both cohorts contribute to a more robust extrapolation of findings to the wider AD population.

Minor Comment 4

Page 9, Overlap Between Cohorts in ICC Edges Analysis:
The description of the methods used to assess the overlap in ICC edges between cohorts could benefit from further clarity. Specifically, the criteria used to define ‘good’ ICC scores and to select top edges for inclusion in the analysis are mentioned but not thoroughly explained. Detailed criteria and justification for these methodological choices would enhance the reader's understanding.

Thanks for this comment. We are now expanding on the use of ICC and the inclusion criteria used. Please see our reply to your previous comment “Statistical Analyses and Interpretation of Results” for further details.

Minor Comment 5

Page 12-13, Chi-square Analysis of Network Edges

While the Chi-square analysis is a suitable method for comparing the proportions of edges within vs. between networks, the application of Bonferroni correction for multiple comparisons is mentioned but not detailed. The manuscript would benefit from a more explicit discussion of how many comparisons were made and the resulting significance threshold after correction.

We thank the reviewer for this comment. We added this in the methods section:

Line 796:

We aimed to determine whether there was a significant difference in the distribution of overlapping edges in within-networks vs. between-networks for each group. To achieve this, we conducted a Chi-square test for each network comparing the number of edges in the within- versus the between-functional networks. Significance was Bonferroni

corrected for multiple comparisons: uncorrected p value was multiplied for the number of comparisons, i.e., $8 = 7$ resting state networks + subcortical regions (Fig. 4B).

Reviewer #2 (Remarks to the Author):

This is an interesting original study, conducted by experts in the field, and therefore with exceptional rigor, which uses connectivity-based brain fingerprinting concept to characterize AD.

I have a number of comments that hopefully will help improving the manuscript:

1

Brain fingerprint is a concept and should not be univocally associated with a specific neuroimaging technique. There are many pre-existing works on functional connectivity network-based person distinctiveness studies, notably with EEG, that would deserve to be cited to give a fairer context in the field.

Thanks for this suggestion. We now acknowledge works that explore fingerprints in different modalities:

Line 59:

“...the extraction of “fingerprints” from human brain connectivity data has become a new frontier in neuroscience, well beyond fMRI data. In fact, studies have investigated brain fingerprints in electroencephalogram (EEG^{1,2}) or functional near-infrared spectroscopy (fNIRS¹), and very recently from magnetoencephalography (MEG) in order to investigate neural features of individual differentiation 1–5.”

2

There are already so many studies on connectivity network-based in AD from disparate structural and functional neuroimaging and electrophysiological data, not only fMRI. Correlation analysis, machine learning approaches, allow for example to give insights into individuality, just as the brain fingerprint approach. It's not clear in what, in essence, brain fingerprint brings something new compared to these existing approaches. Is there more accuracy? Is there better prediction? Is there new mechanisms/knowledge that we

did not before? For example, in the abstract authors say "...by showing that functional connectivity profiles maintain their uniqueness, yet go through functional reconfiguration, during cognitive decline". This sentence is expected to be true for any brain network in which some links remain and other reconfigure across disease. Thus, to better appreciate the obtained findings the abstract, discussion and conclusion, should be shortened and reframed to improve clarity.

We thank the reviewer for this suggestion and we have added this paragraph in the discussion.

Line 396:

This study builds upon an existing body of research in AD utilizing various neuroimaging modalities and analytical techniques, including machine learning approaches, giving insights into individuality. The contribution of this work lies in the novel approach we have taken to individual feature extraction within each specific diagnostic group. While machine learning approaches extract features from existing data and diagnostic classifications that allow new patients allocation to a specific diagnostic group, this approach extracts sources of individual variability specific to a diagnostic group that could help in predicting individual clinical features within each diagnostic group. In this respect, the extraction of individualized FC features through our fingerprinting approach could potentially be used for machine learning prediction. The two approaches are therefore complementary and answer different clinical and research questions.

3

I'm not sure I found any correlation/prediction results between brain-fingerprint features and clinical scores (eg, MMSE). Without such analysis, and without its comparison with more standard brain features, it is impossible to evaluate the relevance of this approach for personalized medicine, which is often advocated in the paper.

We thank the reviewer for this comment and we have added this paragraph in the discussion:

Line 430:

The scope of the current work was to investigate the preservation of identifiability and characterize its topological configuration across the different stages of cognitive decline due to AD. This study answers this fundamental question by pioneering the use of fMRI

data from amyloid-positive Alzheimer's patients across various cognitive decline stages, including dementia, and across two independent cohorts, marking a significant advancement in the field. While the ultimate goal is to utilize individual features of FC to predict behavioral and clinical outcomes, such as cognitive scores, this was not possible in this current work for the following reasons. First, individual variations in clinical features may not be fully characterized by global assessments like the MMSE, which was the only behavioral score consistently available across all subjects in the two datasets. Additionally, MMSE is used as a criterion for patient stratification (e.g., CU: $MMSE \geq 27$; MCI: $MMSE \geq 19$), therefore attempting to predict MMSE scores from FC individual variability within specific patient groups risks circular reasoning and may be confounded by the limited variance observed in MMSE scores within each group (e.g., in ADNI: CU A- 29.2 (SD 0.9); MCI A+ 27.2 (SD 2.4); Dementia A+ 21.6 (SD 2.7)), thereby compromising statistical power. Future work should focus on predicting clinical scores utilizing more sensitive neuropsychological assessments beyond MMSE.

4

In conclusion, the overall length of the text, the resulting redundancy and the adopted phrasing makes difficult to appreciate the real value of the obtained results. I advise the authors to clarify the novelty of their results in a concise way and adopt a more coherent writing style.

We thank the reviewer for this comment. We have substantially reduced the length of the manuscript going from ~6000 to ~4700 for Introduction, Results and Discussion, also in compliance with the journal editorial policies.

REVIEWERS' COMMENTS:

Reviewer #2 (Remarks to the Author):

The authors have addressed all the raised methodological, technical and analysis concerns.

I'm only left with the following sentence "Seminal work in this research area [7-8] has paved the way towards the new promising avenue of detecting individual differences through brain connectivity features », which is not really accurate.

Indeed, the very first work having introduced the idea of brain functional connectivity as biometric trait for person recognition (you can call it brainprint or brain fingerprint) is La Rocca et al, IEEE TBME published in April 2014 (10.1109/TBME.2014.2317881).

Both [7] Finn (2015) and [8] Miranda-Dominguez (November 2014) have been published later.

Including this work would definitely improve the contextualizing introduction and globally acknowledge the efforts done by the community in the field.

Response to reviewers

Reviewers' comments:

Reviewer #1 (Remarks to the Author):

1

The authors have addressed all the raised methodological, technical and analysis concerns.

I'm only left with the following sentence "Seminal work in this research area [7-8] has paved the way towards the new promising avenue of detecting individual differences through brain connectivity features », which is not really accurate.

Indeed, the very first work having introduced the idea of brain functional connectivity as biometric trait for person recognition (you can call it brainprint or brain fingerprint) is La Rocca et al, IEEE TBME published in April 2014 (10.1109/TBME.2014.2317881).

Both [7] Finn (2015) and [8] Miranda-Dominguez (November 2014) have been published later.

Including this work would definitely improve the contextualizing introduction and globally acknowledge the efforts done by the community in the field.

We have revised the manuscript to include the pioneering study by La Rocca et al. (2014). We thank the reviewer for highlighting this seminal work, this addition enhances the contextual framework of our introduction and acknowledges the foundational contributions of the community in this research area.

See Line 52.